# Patient-derived xenografts and organoids model therapy response in prostate cancer

Sofia Karkampouna [1,21], Federico La Manna [1,2,21], Andrej Benjak[3], Mirjam Kiener[1], Marta De Menna[1,4], Eugenio Zoni[1], Joël Grosjean[1], Irena Klima[1], Andrea Garofoli[5], Marco Bolis[6,7,8], Arianna Vallerga[6,7], Jean-Philippe Theurillat[6], Maria R. De Filippo[1,5], Vera Genitsch[9], David Keller[10], Tijmen H. Booij [10], Christian U. Stirnimann[10], Kenneth Eng[11], Andrea Sboner[12], Charlotte K. Y. Ng [3], Salvatore Piscuoglio [5,13,14], Peter C. Gray[15], Martin Spahn[16,17], Mark A. Rubin [18,19], George N. Thalmann[1,20] & Marianna Kruithof-de Julio [1,4,19,20✉]

Therapy resistance and metastatic processes in prostate cancer (PCa) remain undefined, due to lack of experimental models that mimic different disease stages. We describe an androgen-dependent PCa patient-derived xenograft (PDX) model from treatment-naïve, soft tissue metastasis (PNPCa). RNA and whole-exome sequencing of the PDX tissue and organoids confirmed transcriptomic and genomic similarity to primary tumor. PNPCa harbors *BRCA2* and *CHD1* somatic mutations, shows an *SPOP/FOXA1*-like transcriptomic signature and microsatellite instability, which occurs in 3% of advanced PCa and has never been modeled in vivo. Comparison of the treatment-naïve PNPCa with additional metastatic PDXs (BM18, LAPC9), in a medium-throughput organoid screen of FDA-approved compounds, revealed differential drug sensitivities. Multikinase inhibitors (ponatinib, sunitinib, sorafenib) were broadly effective on all PDX- and patient-derived organoids from advanced cases with acquired resistance to standard-of-care compounds. This proof-of-principle study may provide a preclinical tool to screen drug responses to standard-of-care and newly identified, repurposed compounds.

---

[1] Department for BioMedical Research, Urology Research Laboratory, University of Bern, Bern, Switzerland. [2] Department of Urology, Leiden University Medical Center, Leiden, the Netherlands. [3] Department for BioMedical Research, Oncogenomics Laboratory, University of Bern, Bern, Switzerland. [4] Translational Organoid Models, Department for BioMedical Research, University of Bern, Bern, Switzerland. [5] Institute of Pathology, University Hospital Basel, University of Basel, Basel, Switzerland. [6] Institute of Oncology Research, Università della Svizzera italiana, Bellinzona, Switzerland. [7] Swiss Institute of Bioinformatics, Bioinformatics Core Unit, Bellinzona, TI 6500, Switzerland. [8] Molecular Pharmacology and Biochemistry, Mario Negri Institute for Pharmacological Research, Milano 20156, Italy. [9] Institute of Pathology, University of Bern, Bern, Switzerland. [10] NEXUS Personalized Health Technologies, ETH Zurich, Zurich, Switzerland. [11] Department of Physiology and Biophysics, Institute for Precision Medicine and Englander Institute for Precision Medicine, Weill Cornell Medicine,  New York, USA. [12] Department of Pathology and Laboratory Medicine, Institute for Computational Biomedicine, Englander Institute for Precision Medicine, Weill Cornell Medicine, New York, USA. [13] Visceral surgery research laboratory, Clarunis, Department of Biomedicine, University of Basel, Basel, Switzerland. [14] Clarunis Universitäres Bauchzentrum Basel, Basel, Switzerland. [15] The Salk Institute for Biological Studies, La Jolla, CA 92037, USA. [16] Lindenhofspital, Bern, Switzerland. [17] Department of Urology, University Hospital Essen,  University Duisburg-Essen, Essen, Germany. [18] Department for BioMedical Research, Precision Oncology Laboratory, University of Bern, Bern, Switzerland. [19] Bern Center for Precision Medicine, University of Bern and Inselspital, Bern, Switzerland. [20] Department of Urology, Inselspital, Bern University Hospital, Bern, Switzerland. [21]These authors contributed equally: Sofia Karkampouna, Federico La Manna. ✉email: marianna.kruithofdejulio@dbmr.unibe.ch

Prostate cancer (PCa) is the second most commonly diagnosed cancer type and the fifth leading cause of cancer death in men worldwide[1]. Androgen deprivation therapy has been used to hamper tumor growth due to the hormone sensitivity of the prostate. However, a subset of tumors will acquire resistance and reoccur as castration-resistant prostate cancer (CRPC). Novel classes of androgen inhibitors, such as enzalutamide[2] and abiraterone[3], are used for CRPC cases, however acquisition of resistance and intratumor heterogeneity limits their efficiency, thus compelling the use of drugs with different mechanisms of action. The lack of available experimental models of early stage, treatment-naive PCa is a major restriction in preclinical PCa research.

Patient-derived xenografts (PDX) are used to address intra-tumor characteristics and drug response since they model the original tumor in a more representative manner than other models such as two-dimensional cell culture[4]. Various PDX study programs have evaluated the tumor take of various PDX models of primary and metastatic PCa[5–7], with the use of different immunocompromised mouse strains, sites of implantations (subrenal[8], subcutaneous[9], orthotopic[5], intrafemoral[10]) and grafted material (biopsies, cells, circulatory tumor cells[11] and patient-derived organoids (PDOs)[9,12]). Cohorts of available PCa PDXs have expanded in recent years[13–15]. Overall, tumor take proved to be consistently higher for PDXs from advanced and therapy-resistant metastatic cases than for primary PCa, thus underrepresenting early stage cases that can reflect disease progression events[5,16,17].

Tumor-derived organoids recapitulate features of naturally occurring tumors such as cellular phenotype, heterogeneity, drug response, and overall complexity more efficiently than 2D cell lines, while providing an alternative to animal models[18,19]. Several studies have demonstrated that drug response in organoids correlates with concomitant genomic profile and may predict clinical outcome[9,20,21]. Large scale drug screening on primary PCa organoids have not been yet performed, in contrast to CRPC neuroendocrine PCa[12], mainly due to the low proliferation rate of PCa organoids and limited availability of material (e.g. needle biopsies). The extent to which PCa PDX and organoids can model key features of therapy resistance and drug response remains unclear.

In this study, we describe the development of a PDX model derived from a treatment-naïve soft tissue metastasis (PNPCa), with androgen-sensitive characteristics. Molecular characterization revealed distinctive genomic features including *CHD1* and *BRCA2* mutations as well as high microsatellite instability (MSI-H). To assess whether therapy resistance preexists in this treatment-naïve PCa case, we developed a method for organoid derivation that facilitates in vitro immunological assays and drug screening. Using PDX organoids from three different models, we established a pipeline for medium-throughput organoid drug screen and implemented the use of a clinically relevant, near-patient in vitro tool.

## Results

**Establishment of clinically relevant models for human PCa.** We have generated a PDX model from soft tissue metastasis maintained in immunodeficient mice (PNmet) (Fig. 1a). Tissue morphology (Fig. 1b, Supplementary Fig. 1a), presence of luminal markers PSA, NKX3.1, AR, CK8 expression, and absence of CK5+ basal cells (Fig. 1b, Supplementary Fig. 1b–d) indicated stable luminal epithelial morphology among the primary TUR-P tumor (T1), the PNmet, and PDX1–6 passages. Presence of prostatic ducts was observed in the primary T1, PNmet, and PDXs, with a more distinctive ductal morphology in the PDXs (Fig. 1b, Supplementary

Fig. 1a). The PDX was established in two different immunocompromised strains, with and without testosterone supplementation (Supplementary Fig. 1e). Flow cytometric analysis showed that PNPCa PDX cells are positive for prostate-specific membrane antigen (PSMA), E-cadherin (E-Cad), and Integrin α-6 (CD49f), supporting the epithelial and prostatic origin of the tissue. In addition, 37% of cells were CD44+, while a minor fraction stained positively for CD36 (6%) and CD146 (1%) (Fig. 1c).

Tumor growth properties in response to androgen levels, were addressed in vivo. The PDX (passage 6) was implanted subcutaneously and received weekly testosterone injections, reaching ~1000 mm$^3$ volume by day 67 (Fig. 1d, *Intact + Testost*, black line). Surgical castration resulted in progressive tumor regression, reaching non-palpable tumors by day 151 (Fig. 1d, *Castration*, red line). Spontaneous, androgen-independent tumor regrowth was not detected in castrated mice (up to 206 days post-castration, Fig. 1d, day 273), including a setting of prolonged androgen deprivation (up to 289 days post-castration; Supplementary Fig. 2a, *Prolonged Castrated + testost*, blue line). Testosterone supplementation consistently induced tumor regrowth in all castrated groups, with statistically significant tumor burden after 42 days (Fig. 1d, *Castrated + Testosterone*, green line; day 231 $p = 0.0005$ (***), day 238 onwards $p < 0.0001$ (****); Supplementary Table 1) and after 49 days (Supplementary Fig. 2a, *Prolonged Castrated + Testost*, day 392 and 399 $p = 0.0161$, day 403 $p = 0.0018$) of androgen replacement, in the castrated and prolonged castration groups, respectively.

At endpoint, tumors were subjected to histological and transcriptomic analysis. Castration-induced reduction of epithelial NKX3.1+, CK8+, AR+ glands (Fig. 1e; Castrated), which was reversible upon testosterone administration (Fig. 1e; *Castrated + Testost*). Castrated and androgen-replaced tumors were genomically homogeneous as they shared the majority of mutations with the intact tumors (Fig. 1f, Supplementary Data 1). Whereas, after prolonged castration, testosterone-replaced tumors acquired additional mutations in *JAK1* (p.Ala723Ser), *AR* (p.Thr878Ala), and *RET* (p.Ser891Leu) genes (Supplementary Fig. 2b, Supplementary Data 1). Analysis of different organs (liver, lung, prostate, lymph node, femur, and tibia) at end point indicated macroscopic foci on all lung tissues (Supplementary Fig. 2c), loss of normal epithelial architecture in anterior prostate (Supplementary Fig. 2d), human panCK cell infiltration of a lymph node in one case (Supplementary Fig. 2d) and potential micrometastases areas of scattered panCK+ (human specific) cells residing in the bone (Supplementary Fig. 2e), although no apparent lesion was detectable by X-ray (Supplementary Fig. 2f).

Principal component analysis (PCA) and unsupervised hierarchical cluster analysis of RNASeq data (Fig. 1g and Supplementary Fig. 3a) indicated high transcriptomic correlation among tumors from intact and testosterone-replaced hosts, while tumors from castrated hosts further diverged from the aforementioned groups (Fig. 1g). Differential expression analysis revealed lower expression of genes in metabolic pathways, mTOR, MYC, and AR pathways in the Castrated group compared to the Intact tumor groups (Fig. 1h). In particular, cluster analysis of AR pathway showed the differential expression of multiple genes in the castrated group, compared to both the intact and the androgen-replaced groups (Supplementary Fig. 3b). The latter groups shared similar transcriptomic profile and androgen sensitivity, however, pathway analysis indicated that tumors from androgen-replaced hosts had significant downregulation of cell cycle, apoptosis and hypoxia-related pathways (NES ≤ −1), and upregulation of interferon response, protein secretion and androgen response pathways (NES ≥ 1) (Supplementary Fig. 3c).

Androgen ablation by castration caused tumor regression, whereas testosterone re-administration induced tumor regrowth,

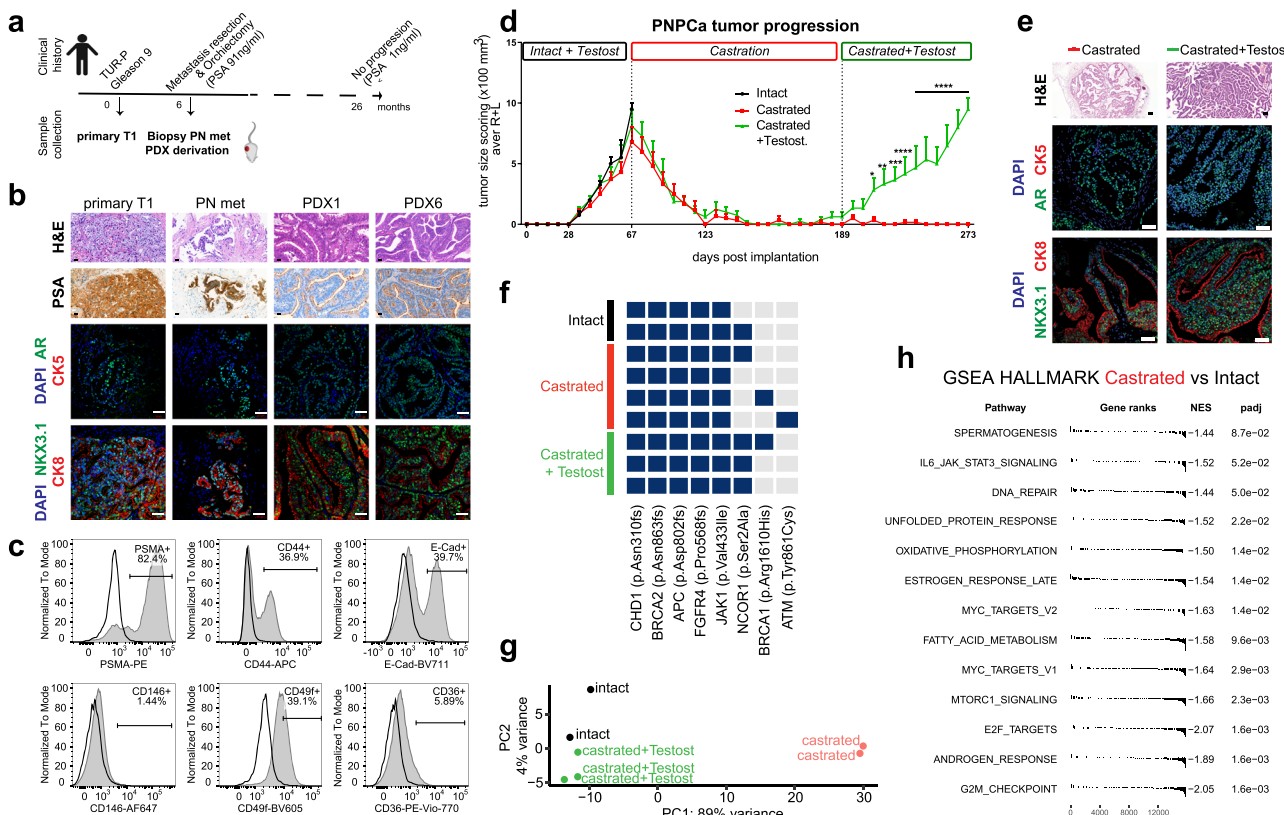

**Fig. 1 Establishment of a novel androgen dependent, patient-derived xenograft from an early, treatment-naïve prostate cancer metastasis. a** Scheme of clinical history and patient-derived samples: primary tumor TUR-P (T1) and penile metastasis needle biopsies used to establish the PDX model (PNPCa) and subsequent passages. Created with BioRender.com. **b** Histological morphology of primary TURP tumor, penile metastasis (PN met) from PCa and the PDX passages 1 and 6 (PDX1, PDX6) derived from the metastasis needle biopsy implantation (PNPCa), as assessed by Hematoxylin and Eosin staining (H&E). Scale bars 20 μm. *Top to bottom panels*: PSA protein expression. Scale bars 20 μm. Expression of AR (green), CK5 (red) assessed by immunofluorescence, DAPI (blue) marks the nuclei. Scale bars 50 μm. Expression of NKX3.1 (green), CK8 (red) assessed by immunofluorescence. Scale bars 50 μm. **c** Flow cytometry analysis of epithelial and prostate-specific marker expression in PNPCa PDX tissue. FcR-blocked PNPCa cells were stained with antibodies against CD44, E-Cadherin, PSMA, CD49f, CD36, and CD146. **d** PDX tumor growth progression in time. Groups; 1. Intact tumors (collected at max size, $N = 2$ independent animals), 2. Castrated ($N = 5$ independent animals), 3. Castrated followed by Testosterone re-administration (Castrated-Testosterone independent animals) starting on day 189 ($N = 4$, $N = 3$ from day 203 to 252, $N = 2$ from day 252 to 273). Tumor scoring was performed weekly by routine palpation; values represent average calculation of the tumors of all animals per group (considering $N = 2$ tumors, left L and right R of each animal). Error bars represent SEM, calculated considering number of animals for each time point. Ordinary two-way ANOVA with Tukey's multiple comparison correction was performed. (*) $p = 0.0105$ day 217, (**) $p = 0.0025$ day 224, (***) $p = 0.0005$ day 231, (****) $p \leq 0.0001$ from day 238 onwards. **e** *Top to bottom panels*: Histological H&E staining of representative tumors from Castrated and Castrated-Testosterone hosts. Immunofluorescence staining for AR and CK5, CK8, NKX3.1 and CK8, CD44, and Ki67. DAPI marks the nuclei. Scale bars 50 μm. **f** Genomic analysis of PNPCa PDXs from intact and castrated animals, collecting samples at full regression (122 days) and after further testosterone replacement (84 days). **g** Principal component analysis of the gene expression of the 500 most variable genes. **h** Gene set enrichment analysis plot of statistically significant (adjusted *p*-value < 0.05) enrichment of HALLMARK pathways based on the differential expression analysis of the Castrated versus the Intact groups. NES, normalized enrichment score.

rendering the PNPCa an androgen-dependent, treatment-naïve model.

**Molecular analysis revealed genomic and transcriptomic stability among the PCa xenograft and organoid models.** To further explore the in vitro characteristics of the PDX tumor cells, we developed an organoid culture method from bulk tumor tissue that allows organoids to grow in suspension conditions, with no requirement for extracellular matrix support (e.g. Matrigel). Two previously established bone-metastatic PCa PDX models, BM18[22] and LAPC9[23] were used for direct comparison with the PNPCa, representing respectively, androgen-sensitive and androgen-independent, advanced PCa models (Supplementary Fig. 4a, b). Organoids derived from all investigated PDXs displayed budding acinar and adenocarcinoma-like morphology, with the expression of the luminal markers CK8, PSA and AR (Fig. 2a). Gene

expression of basal markers (*CD49f/ITGA6, KRT5, KRT6*) was detected but less abundant than luminal ones (*NKX3.1, AR, CK18*) in all organoid and PDX models (Supplementary Fig. 5a). Tumorigenic potential of BM18 and LAPC9 organoids was confirmed by subrenal grafting (Supplementary Fig. 4c–h) or intraprostatic inoculation (Supplementary Fig. 4i–n). Organoid viability was enhanced when cultured in dihydrotestosterone (DHT)-supplemented organoid media (Fig. 2b).

Next, we analyzed the transcriptomic and genomic profiles of PDX organoids and matched tissues, assessing its stability across multiple samples (Supplementary Fig. 5b). PCA indicated that the three PDX models were clearly distinct, while the matched PDX tissues and organoids were highly similar (Fig. 2c). This was further demonstrated by the high gene expression correlation scores in PDX tissues and organoids, generated using the developed culture system (Fig. 2d).

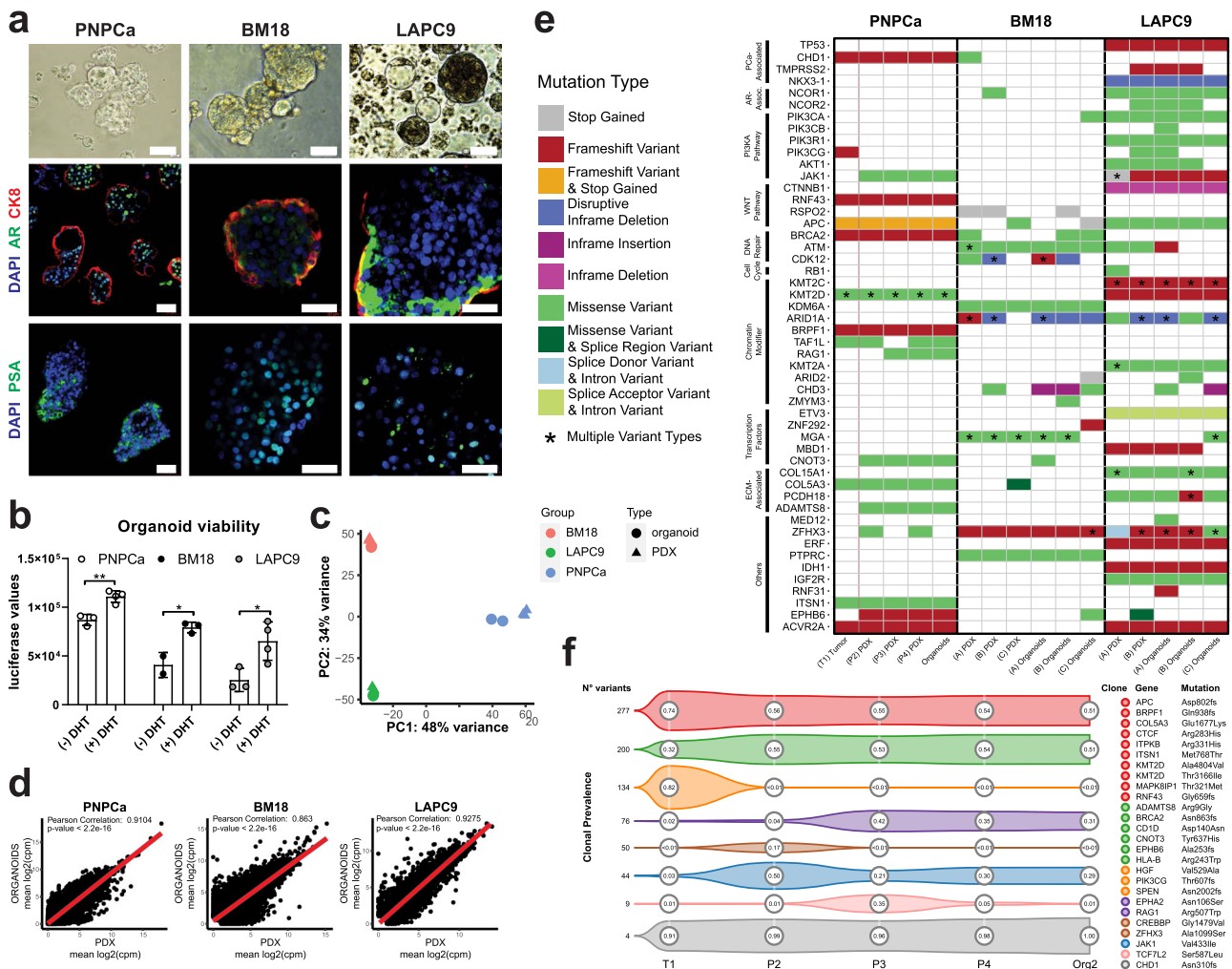

**Fig. 2 Mutational landscape of PDX and PDX-derived organoids from PNPCa, and advanced androgen (in) dependent BM18 and LAPC9 models.**
**a** Morphology of PNPCa, BM18, and LAPC9 PDX-derived organoids; brightfield images, whole mount immunofluorescence staining and 3D projection of z-stack of organoids stained for PSA, AR, and CK8. DAPI marks the nuclei. Scale bars 50 μm. **b** Viability assay of organoids derived from PNPCa, BM18, and LAPC9 tumor tissues and exposed to dihydrotestosterone (±DHT) for 48 h. Luciferase values (ATP release) are proportional to cell viability. Mean ± SD is reported, $N = 3,4$ technical replicates per condition (PNPCa), $N = 2,3$ technical replicates per condition (BM18), $N = 3,4$ technical replicates per condition (LAPC9). Two-tailed $t$-test, $*p = 0.0161$, $p = 0.0277$, $**p = 0.0031$. **c** Principal component analysis of the gene expression of the 1000 most variable genes on PNPCa, BM18, and LAPC9 samples (PDX and PDX-derived organoids). **d** Correlation plots of gene expression between PNPCa PDX tissue ($N = 3$ biologically independent tumor samples) and organoids ($N = 2$ biologically independent organoid samples), BM18 PDX tissue ($N = 2$) and organoids ($N = 2$), LAPC9 PDX tissue ($N = 2$) and organoids ($N = 2$), $p$-values < $2.2 \times 10^{-16}$. **e** Somatic mutation analysis of WES of tissue and organoids of PNPCa, BM18, and LAPC9 PDX. Columns represent different samples, while rows represent selected genes categorized by pathway. Types of genetic aberrations are indicated in different colors. Multiple types of mutations per gene are indicated with an asterisk. A–C indicate biological replicates. **f** Clonality analysis of the PNPCa samples shown in **e**, inferred by PyClone. Only the largest clones (consisting of most variants) or those containing cancer genes are shown. Numbers in circles indicate mean clonal prevalence, estimated for each sample. Mutations in cancer genes corresponding to each clone are reported on the left and color-coded. Overall, most mutations (including those in cancer genes) occur at high prevalence in all samples (top two clones). WES, whole-exome sequencing.

We next performed gene set enrichment analysis (GSEA) using Hallmark and KEGG (C2 subset) gene sets (Supplementary Fig. 6, $\log_2 FC$ (NES) ≥ | 1 |, FDR > 0.05). In all samples compared, we found a significant upregulation of pathways involved in cell growth (E2F targets, G2M checkpoint, DNA repair mechanisms, MYC targets), metabolic activity (amino acid and fatty acid metabolism, oxidative phosphorylation, protein secretion) and androgen-regulated pathways (androgen response, spermatogenesis). Downregulated pathways included epithelial plasticity (epithelial–mesenchymal transition, adhesion molecules regulation), angiogenesis (angiogenesis, coagulation) and cellular interactions with immune system (antigen presentation, interferons, and cytokines pathways).

We then assessed the genomic landscape of PNPCa PDX model by whole exome sequencing (WES) (Fig. 2e, Supplementary Figs. 5b, 7a, b, Supplementary Data 2, Supplementary Table 2). Copy-number analysis showed that PDX-derived organoids retained the overall patterns of copy number alterations (CNAs) of tissues (Supplementary Fig. 8a). Of the total somatic non-synonymous mutations found, 51% were overlapping between all samples (Supplementary Fig. 8b, Supplementary Data 2). WES was additionally performed on the LAPC9 and BM18 PDXs and organoids, which recapitulated the majority of the somatic mutations detected in the matched PDX tissues and organoids (Fig. 2e, Supplementary Fig. 7c, d, Supplementary

Data 2, Supplementary Table 2). Frameshift mutations of high impact detected in the BM18 model were in *ATM*, *ZHFX3,* and in the chromatin modifiers *KDM6A* and *ARID1A* genes (Fig. 2e). LAPC9 PDX returned an overall higher mutational burden, which included a *NKX3.1* deletion, along with frameshift mutations in *TP53*, *TMPRSS2*, and *PI3K* and in genes of the Wnt pathway (*APC*, *CTNNB1*) (Fig. 2e). Nearly all non-synonymous somatic mutations in bona fide cancer genes were preserved in the PNPCa models, including truncating mutations in *CHD1*, *ACVR2A*, *RNF43*, *APC*, and *BRCA2* (Fig. 2e, Supplementary Fig. 7a). Three frameshift mutations in cancer genes were lost in the PDX tumors: *HGF*, *SPEN*, and *PIK3CG* (Fig. 2e, Supplementary Fig. 7a). No mutations in the *AR* gene were identified, however mutations in the AR pathway-associated gene *UBE3A* were detected, while mutations in *PMEPA1* and *KDM3A* were observed sporadically (Supplementary Fig. 7b). We determined the cancer cell fraction (CCF) of the reported mutations to determine their stability over PDX passaging and organoid generation. Mutations with homogeneously high prevalence in all samples (≥80% CCF) were found in *CHD1*, *APC*, *RNF43*, and *KMT2D* (Supplementary Fig. 9). *BRCA2* and *ACVR2A* mutations were found in 60–90% CCF in P2, P3, P4, and Org2 and in ≤20% CCF of T1 (Supplementary Fig. 9). We next analysed the mutational variation of the sequenced PNPCa samples by performing clonal analysis. Different clone dynamics were detected for the most abundant clones and are reported in Fig. 2f. Of relevance, *CHD1* mutation was shared among all subclones while two main subclones captured discrete subsets of truncal mutations from the original primary tumor (T1) and stably transmitted across PDX passages and organoid samples.

We compared gene expression of PNPCa samples to that of genetically defined subgroups within the Cancer Genome Atlas (TCGA). The PCA plot reports the variance of PCa cases with *CHD1* homozygous deletion (Supplementary Fig. 10a) or with mutant *FOXA1*, *SPOP*, *CHD1*, ETS rearrangements (*ERG*, *ETV1*, *ETV4*) (Supplementary Fig. 10b). We further cross-compared those samples for their expression of genesets specific of different signatures of PCa subtypes using single-sample gene-set enrichment analysis (ssGSEA). While the signatures of ETS and SPOP/FOXA1 subgroups were divergent from each other, all PNPCa samples, clustered between the two categories (Supplementary Fig. 10c) and closely to CHD1 homozygous-deletion group (Supplementary Fig. 10a), in agreement with presence of a truncating *CHD1* mutation (Fig. 2d).

Additionally, we used transcriptomic analysis to compare the PNPCa PDX to a subcohort of the LuCaP PDXs series[13]. Unsupervised hierarchical cluster analysis placed PNPCa closely with LuCaP-78 which, despite being classified as androgen-sensitive adenocarcinoma-like PCa, was derived from a CRPC patient already exposed to different lines of treatment, including taxanes (Supplementary Fig. 10d). From a genomic standpoint, LuCaP-23.1 and 145.2 contain heterozygous loss of *BRCA2*, a trait associated with progression to CRPC[24], while only LuCaP-147 is reported to have a hypermutation profile associated to MSI-H[25]. Thus, PNPCa PDX represents a model of treatment-naïve, early-stage advanced disease, recapitulating features commonly found in more progressed stages of PCa.

**Functional testing of targeted treatments according to the genomic profile of the PNPCa PDX and organoids**. We next characterized the functional implications of mutations in genes linked to DNA stability, detected in the PNPCa models. In vitro viability assays on irradiated PNPCa organoids showed a significant reduction of viability already after 48 h compared to control PNPCa organoids (Fig. 3a). Conversely, no difference in

viability was evident in irradiated organoids derived from BM18 or LAPC9, confirming the sensitivity of PNPCa to irradiation in this experimental setup (Fig. 3a–c).

The mutational landscape of PNPCa primary tumor, PDX and organoids was largely driven by mutational processes associated with signatures of MSI (Fig. 3d), consistent with the observations that this tumor had a higher mutation rate compared to other PCa[26], especially to those with an overall flat copy number profile (Supplementary Fig. 8a) and an elevated proportion of small indels[27] (Supplementary Fig. 11a). We did not observe genomic markers indicative of homologous recombination deficiency, neither in terms of mutational signature (Fig. 3e) nor of large-scale transitions[28]. The MSI status was further evaluated using the MSIsensor algorithm[29] and the Bethesda MSI test. MSIsensor classified all the samples except the T1 tumor as MSI-H (Fig. 3e, gray dotted line). Four out of the six loci of the Bethesda panel were altered in the T1 tumor, confirming the tumor itself as MSI-H (Supplementary Fig. 11b).

Considering that MSI score/defective mismatch repair (dMMR) mechanism are used as biomarkers for PD-L1/PD-1 immunotherapy response[30,31], and correlate with PD-L1 over-expression[32], we assessed the expression level of immune-related markers and the overall immunomodulatory properties of PNPCa organoids. Compared to the normal tissue N1, all the PDXs, Org2, and the primary tumor showed a reduction of transcript abundance of the major histocompatibility complex (HLA-A and HLA-B) and of galectin-9, the main ligand of the inhibitory receptor Tim-3. While the primary tumor, the soft tissue metastasis, and the PDX-derived organoids showed a moderate, epithelial-specific staining for PD-L1, PDX1, and PDX2 tissues revealed a loss of expression of this marker in the epithelial compartment (Fig. 3f). The increased level of expression of PD-L1 in the primary tumor compared to healthy tissue N1 was not preserved in the PDX and in the organoid samples (Supplementary Fig. 11c, d).

We further assessed the expression of immune checkpoint inhibitors after 48 h stimulation of PNPCa organoids with 50 ng/ml IFN-γ. Treatment with IFN-γ was sufficient to significantly evoke the upregulation of PD-L1 ($p < 0.001$) and galectin-9 ($p = 0.0011$), linked to immune evasion (Fig. 3g, Supplementary Fig. 11e)[33,34]. However, as the molecular and histological data could not provide a unitary picture for these markers, we further investigated their functional role by co-culturing PNPCa organoids with allogeneic CD3+ lymphocytes and mature dendritic cells (mDC). Despite the upregulation of immune checkpoint inhibitors at the molecular level, PNPCa organoids did not modulate the proliferation, T$_{reg}$ polarization or PD-1 expression of CD3+ lymphocytes, even after 48 h pre-treatment with IFN-γ (Fig. 3h–j). Overall, the current results do not support the presence of an active immune evasion machinery in PNPCa organoids.

**Organoid drug response to standard-of-care and repurposing of FDA-approved compounds on a medium-throughput automated screen**. We next implemented the PDX-derived PCa organoid in a medium-throughput automated drug screening pipeline. A total of 74 compounds, including routinely used PCa standard-of-care drugs as well as different FDA-approved drugs with indications for other cancer types were assayed, at multiple concentrations (Supplementary Data 3). Overall, the tested compounds targeted several distinct cellular processes and pathways, with a specific focus on signaling mediators including growth factor receptors and androgen response.

Pre-screens were run for every PDX-derived organoid model to optimize cell density, positive controls and time of drug exposure,

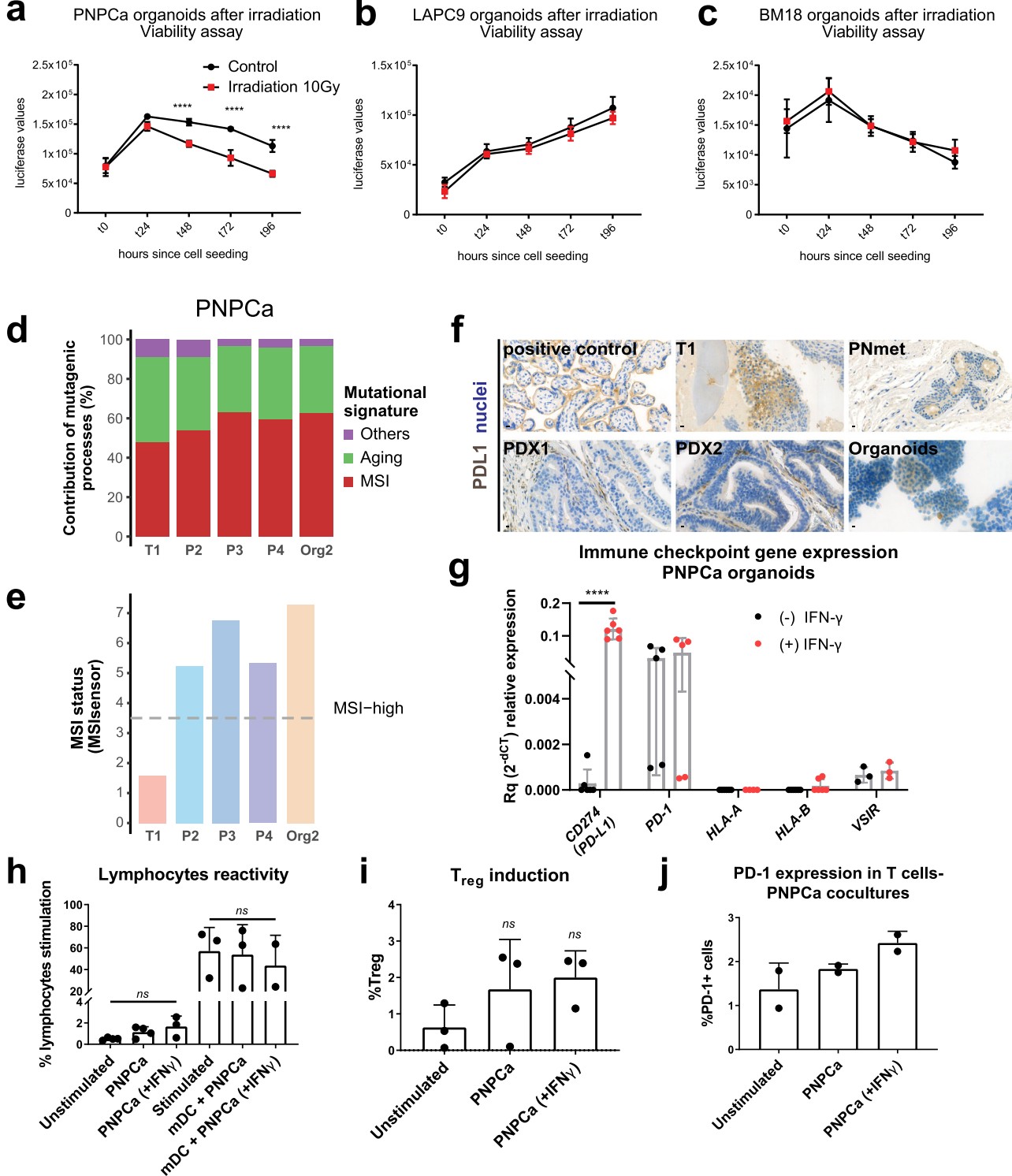

according to a pipeline established using PNPCa organoids (Supplementary Fig. 12a, b). Screens were run in at least three independent replicates for each PDX model, validating each replicate before data integration (Supplementary Fig. 12c, d). Once automatically seeded, organoids were allowed to form for 48 h before adding the drugs; cell viability was assessed on-plate after 48–72 h from initial drugs exposure (Fig. 4a).

PNPCa organoids showed resistance to the majority of the tested drugs, however 14 drug compounds significantly reduced their viability (Fig. 4b, Table 1, FDR < 0.05). The most effective compounds targeted the AR pathway (enzalutamide), EGFR/

HER2 (afatinib, erlotinib, gefitinib), mTOR (rapamycin, temsirolimus), DNA replication (doxorubicin, daunorubicin, and epirubicin), multiple tyrosine kinase pathways (sorafenib, ponatinib, sunitinib), c-Met (crizotinib), and Src (bosutinib). Performing the drug screen on multiple PCa PDX-derived organoid models identified drugs that were both exclusively effective in only one of the models as well as drugs broadly effective across all tested PDX-derived organoids (Fig. 4b, Supplementary Fig. 13). As expected, the amount of significant hits inversely correlated with tumor aggressiveness. In particular, the androgen-sensitive BM18 showed a specific sensitivity to taxanes, while the

**Fig. 3 Correlation of genomic features and specific drug responses in organoid models. a–c** Time course of ATP-mediated luminescence viability assay following a single dose of 10 Gy irradiation on organoids derived from PNPCa (**a**), LAPC9 (**b**), and BM18 (**c**) PDX tumors. Mean ± SD is reported, $N = 4$ technical replicates ($t = 0$), $N = 5$ (for each of the $t = 24$, 48, 72, 96 h time points). Ordinary two-way ANOVA with Tukey multiple comparison test was performed. ****$p < 0.0001$. **d** Graph representing the percentage of contribution of specific mutagenic processes based on mutational signatures from PNPCa T1 (primary tumor), PDX (passages P2-P4) and organoids (from P4 PDX). **e** MSI status based on MSIsensor algorithm (https://github.com/ding-lab/msisensor), score ≥ 3.5 indicates MSI-high. **f** PD-L1 IHC staining on positive control (placenta tissue), primary T1 tumor, PNmet needle biopsy, PDX1 and PDX2 of the PNmet, and cytosmear of PDX-organoids. Images of representative areas per tumor sample are shown, relative to the positive control staining. **g**. Gene expression levels of immune markers based on RT-qPCR results on PNPCa organoids RNA at baseline (black bars) and after 48 h exposure to IFN-γ (red bars). Mean ± SD is reported, for *VSIR* $N = 3$, for *PD-L1* $N = 6$, for *PD-1*, *HLA-A*, *HLA-B* $N = 5$ technical replicates, across two independent experiments. Two-tailed nested *t*-test. ****$p < 0.0001$. **h–j** MLR assay showing lymphocyte reactivity, $T_{reg}$ fraction and expression levels of surface PD-1, following coculture of PDX-derived PNPCa organoids with T cells and allogeneic, monocyte-derived dendritic cells (DCs). Mean ± SD is reported, **h** $N \geq 3$ biologically independent samples per condition, $N = 2$ mDC+ IFN-γ from four independent experiments; Mixed effects analysis (REML) with Geisser–Greenhouse's correction and Dunnett's post-hoc test was performed. **i** $N = 3$ biologically independent experiments; one-way ANOVA with Dunnett's correction for multiple comparisons was performed, **j** $N = 2$ per condition from two biologically independent experiments. IHC, immunohistochemistry; MLR, mixed lymphocyte reaction.

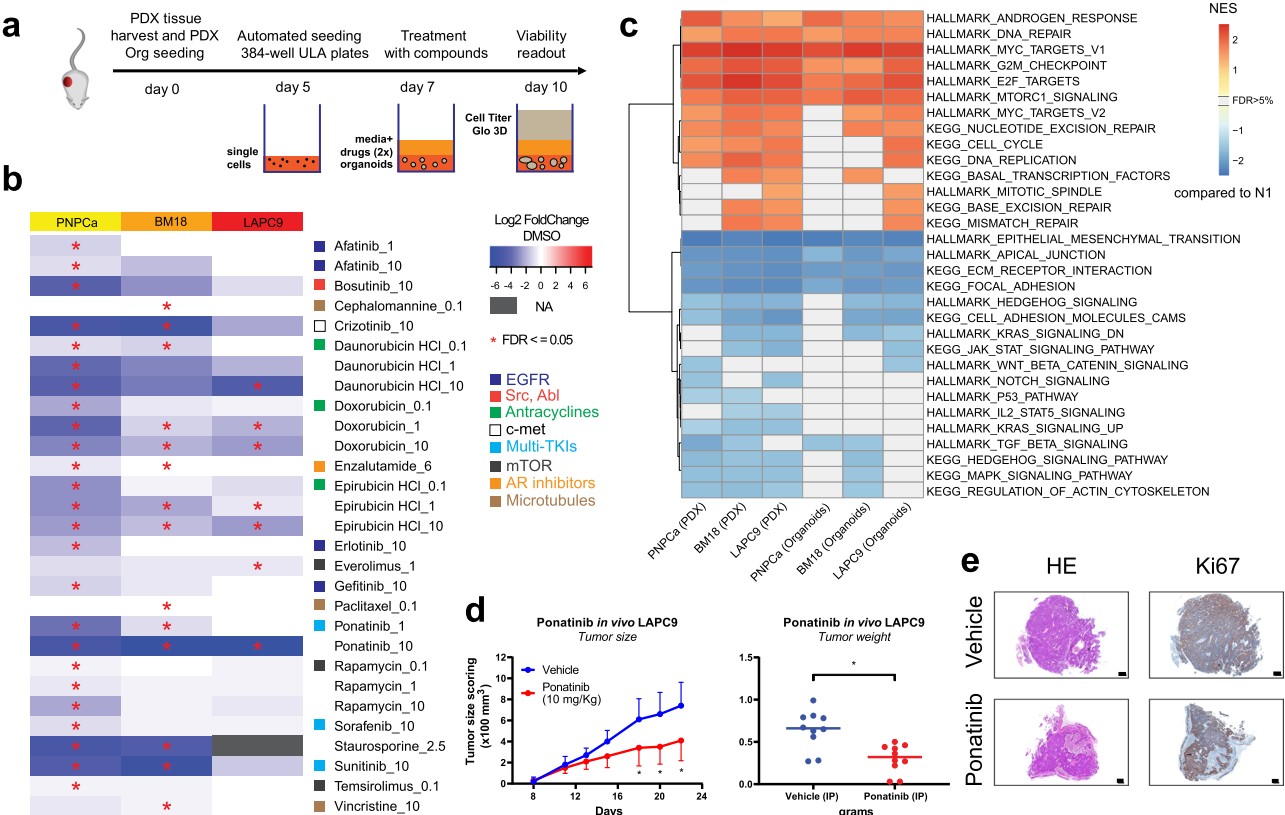

**Fig. 4 Drug sensitivity of organoids representing different PCa stages and identification of novel compounds for repurposing use, based on medium-throughput organoid screens. a** Scheme of experimental protocol for organoid drug screens. Created with BioRender.com. **b** Organoid drug screen heatmap of $\log_2$ fold change viability values (over vehicle, for each PDX model) for PNPCa ($N = 4$ replicates), BM18 ($N = 3$), and LAPC9 ($N = 3$). Negative $\log_2$ values (plotted in blue) indicate potential drug candidates with impact on cell viability. Staurosporine was used as positive control. Statistically significant hits (FDR ≤ 0.05) are indicated with an asterisk. Hits with a significant effect on at least one model are reported, listed in alphabetical order and with effective dose indication on the right, in μM. Medium-throughput automated drug screens, using selected FDA-approved compounds, were performed at Nexus Theragnostics platform. **c** Gene set enrichment analysis (GSEA) of PDXs tissue (PNPCa $N = 3$, BM18 $N = 2$, LAPC9 $N = 2$) and organoids (PNPCa $N = 2$, BM18 $N = 2$, LAPC9 $N = 2$). Enrichment scores of selected Hallmark and KEGG (C2) pathways with FDR < 0.05 derived from differential expression analysis of each group of samples vs. the non-carcinoma control tissue from PNPCa clinical sample (N1). NES normalized enrichment score. **d** In vivo efficacy by ponatinib treatment in subcutaneous LAPC9 PDX model. Tumor-bearing mice received intraperitoneally (IP) daily injections of vehicle or ponatinib (10 mg/kg) and mean tumor size scoring (×100 mm³) was plotted. Data are presented as mean ± SD, $N = 10$ independent tumor samples per treatment group; Two-way ANOVA with Sidak correction. *$p = 0.029$ (day 18), $p = 0.012$ (day 20), $p = 0.016$ (day 22). Tumor weight was assessed at endpoint and plotted as mean ± SD, $N = 10$ independent tumor samples per group; Two-tailed nested *t*-test. *$p = 0.015$. **e** Representative histology of LAPC9 PDX tumors from the vehicle and ponatinib groups, collected at endpoint. HE, hematoxylin and eosin, Ki67 proliferation marker. Scale bars, 0.5 mm.

castration-resistant LAPC9 to mTOR inhibition by everolimus. All models were sensitive to the tested anthracyclines (doxorubicin, daunorubicin, and epirubicin) targeting DNA replication and to the tyrosine kinase inhibitor ponatinib.

We next performed targeted pathway analysis to elucidate if the pathways targeted by the screened drugs (e.g. mTOR, cell cycle, AR, TKI targets; EGFR, FGFR, Jak/STAT, Src, PDGFRB) were enriched in the PDX tissues as well as in their derived

organoids. A curated list of Hallmark and KEGG C2 pathways significantly enriched in the PDX tissues and organoids is reported in Fig. 4c. Differential expression and fGSEA analysis indicated a significant upregulation of pathways linked to AR signaling, DNA repair, mTOR signaling, and cell cycle progression across all PDX models and organoids, compared to normal tissue (Fig. 4c, pathways with NES ≥ 1). Overall, pathway analysis supported the results of drug screening on organoids and highlighted the molecular correlation of tissue/organoids on pharmacologically relevant pathways.

In order to validate the drug candidates, we performed drug screens on ex vivo tissue slices from PNPCa, BM18, and LAPC9 PDX. We tested 13 compounds on PNPCa tissue slices including 12 of the effective compounds and docetaxel, which did not significantly affect organoid viability in the PNPCa drug screen. Of these, we were able to validate 11 out of the 13 tested compounds (Supplementay Fig. 14a, b). We then tested a selection of drugs, both effective and ineffective in the organoid screening, on ex vivo tissue slices of LAPC9 and BM18 PDX. We validated the effectiveness of 7 out of 11 tested compounds on LAPC9 PDX and of 4 out of 5 tested compounds on BM18 PDX (Supplementary Fig. 14c). Overall, the effective compounds identified were part of three drug classes: anthracyclines, mTOR inhibitors, and TKIs.

Among the multiple TKIs effective on the different organoid models tested (Fig. 4b), ponatinib showed a clear dose-dependent effect, both in the organoid and in ex vivo tissue slice assay, while it has not been previously studied in the context of PCa. We validated the in vivo efficacy of ponatinib on the most aggressive PDX, LAPC9. The vehicle formulation was optimized for intraperitoneal (IP) injection and matched the solubility profile of vehicle DMSO (Supplementary Fig. 15a, b). Mice treated with ponatinib had a significantly lower tumor burden compared to controls already after 18 days from PDX implantation, resulting in significantly lower tumor weight and volume in the treated group at the end of the experiment (Fig. 4d, Supplementary Fig. 15c, d). Moreover, treatment with ponatinib reduced the average mouse weight loss seen in the control group and likely ascribable to the effects of PDX growth (Supplementary Fig. 15e). Histological morphology and quantification of Ki67 indicated no significant treatment-dependent differences in proliferation rates of the LAPC9 xenografts (Supplementary Fig. 15f, g). The concordance of the in vivo data with the ex vivo and in vitro drug screens is supporting the feasibility of an organoid-based drug assay as a surrogate tool for in vivo response, therefore we further investigated the efficacy of the identified compounds on patient-derived material.

**Defining a drug panel for therapy resistant PCa (PDXs and patient-derived material) for routine organoid screens and treatment decision.** In order to develop a precision medicine approach for PCa patients, we established PDOs from needle biopsies of radical prostatectomy and metastatic specimens. As a control for PDO formation efficiency, we sampled areas macroscopically unaffected by cancer in radical prostatectomy specimens, based on the evaluation of board-certified pathologists. On average, the matched control tissues formed fewer organoids compared to samples from malignant PCa (Fig. 5a, "benign" and "tumor"). PCa PDOs showed two main morphological phenotypes in vitro: organoids with a more acinar or cystic morphology and organoids with an adenocarcinoma-like phenotype (Fig. 5a, "acinar" and "adenocarcinoma"). Although PDO cultures showed both inter- and intra-patient morphological heterogeneity, each PDO culture showed consistent morphology across passages. Tumorigenic potential of PCa organoids was assessed by in vivo

intraprostatic injections (Supplementary Fig. 16, representative cases).

Matched patients' blood, bioptic tissue, and PDOs were subjected to targeted RNA and DNA sequencing using a panel of clinically relevant cancer-related genes and according to specimen abundance (Fig. 5b–d, Supplementary Table 3). The demographic characteristics of this cohort are included in Supplementary Table 4. In 54.5% of cases, the somatic mutations identified in PDOs were matching those identified in the matched tissue (Fig. 5b, purple), in 27.3% of cases no genomic concordance/mutations in the organoids were found (Fig. 5b, dark gray), and in 18.2% of the cases no mutation was identified in both the organoids and the original tissue (Fig. 5b, light gray), possibly due to sampling a no-tumor or low-tumor area of the prostate. A binary heatmap plot details the different mutations detected in the analyzed samples (Fig. 5c). Comparison of transcriptomic profile of PCa bioptic tissues with matched PDOs resulted in high correlation in the cases evaluated (Fig. 5d, cases P61 and P62). Moreover, the analysis confirmed the prevalence of luminal over basal markers in the analyzed PDOs (Supplementary Fig. 17).

Of the effective drugs short-listed from the PDX-derived organoids drug screening, we selected a panel of 13 most effective compounds (based on statistical significance), together with 4 PCa standard-of-care compounds, to develop an in vitro, multi-drug, PDO-based precision medicine assay. PDOs from three advanced PCa cases as well as from two primary PCa cases were screened with the assay (Fig. 5e). Confirmation of tumor content was done either by genomic sequencing (Fig. 5c) or after histopathological evaluation by certified pathologist. Normalized viability scores for each assay are reported on the heatmap in Fig. 5e and unsupervised hierarchical cluster analysis identified compound classes with differential effectiveness. AR-interfering drugs (abiraterone and enzalutamide) and docetaxel clustered together with the controls, indicating overall a low efficacy on PDOs. Drugs targeting mTOR (rapamycin, temsirolimus, everolimus) and EGFR-targeting erlotinib significantly reduced viability of PDOs only in some cases, showing differential efficacy. A third group, consisting of anthracyclines, the mTKIs ponatinib, bosutinib, and sunitinib, together with crizotinib, were broadly effective, reducing PDO viability in most cases. From a clinical standpoint, PDOs from patients receiving in their treatment lines taxanes (case P89) or ADT (cases P80, P82, P89, P133) confirmed resistance to these compounds in vitro with the exception of cases P82 and P134, which were sensitive and resistant, respectively, to enzalutamide despite undergoing (case P89) or a lack of ADT treatment (case P134). Although larger sample size would increase statistical robustness, this proof-of-principle data endorses the applicability of a near-patient PDO-based approach for personalized drug screening for PCa patients.

## Discussion

In this study, we describe the establishment of a PCa PDX model, from an early onset and treatment-naïve patient, representing a critical stage of PCa between initial relapse and CRPC development. We established organoid cultures from this and from more advanced PCa PDXs (BM18, LAPC9), to develop an organoid-based drug screening pipeline. We then further adapted organoid cultures to patient-derived bioptic material, implementing a clinically relevant, patient-tailored organoids drug screening assay.

PCa is generally a slow proliferating tumor with few key mutations and genetic alterations commonly found in patients such as *TMPRSS-ERG, SPOP, FOXA1, PTEN*[35,36]. *SPOP* is associated with DNA repair errors and a higher number of genomic rearrangements[37]. Although no *SPOP* mutation was identified in the PNPCa, there was a higher transcriptomic correlation with

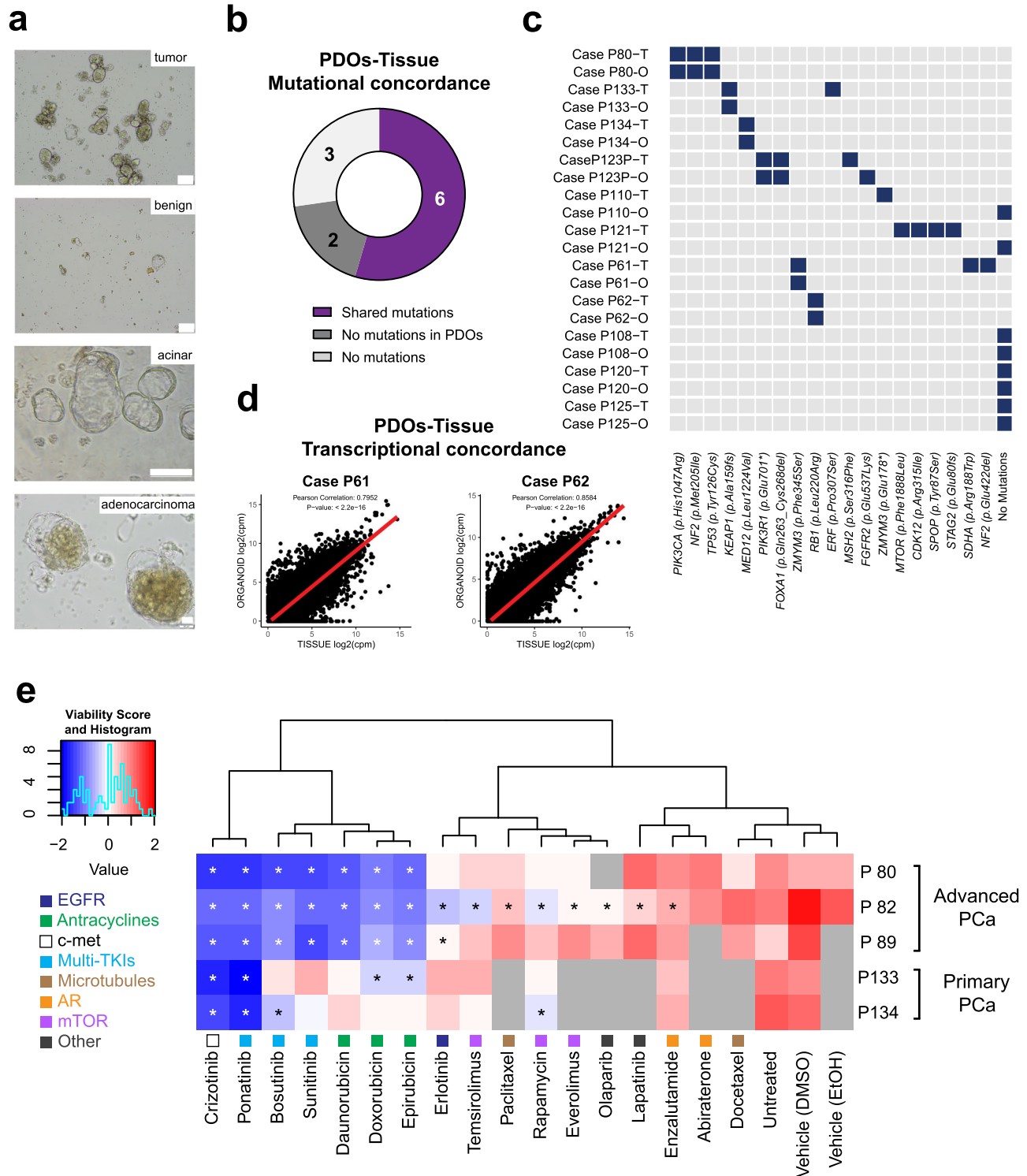

*SPOP*-mutated cases compared to other subtypes and specifically to the *ERG*-mutated ones: as characterized in previous studies, *SPOP* mutations and *ERG* fusion are fundamentally mutually exclusive[38,39]. Nevertheless, PNPCa PDX carries an inactivating mutation in *CHD1*, an event frequently co-occuring with *SPOP* mutations[40] (15% of PCa cases show loss of heterozygosity for *CHD1*) and driving PCa-specific growth in transgenic mice[41].

The PNPCa model displays an heterozygous frameshift mutation in the *BRCA2* gene and high MSI, combining in one model two of the four genomic subtypes of metastatic CRPC[26]. Patients with germline *BRCA2* defects have earlier disease onset, a higher rate of 5-year metastatic progression and poor survival compared to non-carriers[42–44]. PNPCa organoids exhibited sensitivity to irradiation, a clinical treatment for patients harboring *BRCA2* mutations[42]. Tumors with *BRCA1/2* defects and defective DNA repair mechanism are particularly sensitive to DNA damaging agents such as radiotherapy[45] and to PARP inhibitors such as olaparib, in use for breast and ovarian cancer[46] and currently in a phase III clinical trial for CRPC patients harboring *BRCA1/2* or *ATM* mutations (PROfound, NCT02987543).

To our knowledge, PNPCa is the only reported MSI-H case of an early metastasis, retaining androgen sensitivity that has been

**Fig. 5 Patient-derived organoids (PDO) of multiple PCa cases preserve molecular signatures of the matched tissue and can be used to determine drug sensitivity in vitro. a** Representative brightfield images of PDOs from PCa ("tumor") and from cancer unaffected control area ("benign") from the same patient (scale bar 100 μm). Representative images of PDOs with an acinar or cystic morphology and with an adenocarcinoma-like morphology ("acinar" (scale bar 500 μm) and "adenocarcinoma" (scale bar 50 μm), respectively). **b.** Overview of the genetic profiles of $N = 11$ PCa tissue samples and matching PDOs, determined by targeted sequencing of PCa-specific mutation panels. **c** Binary heatmap plot of the data presented in **b**. Rows represent samples (tissue (T) and organoids (O) for each case), columns represent genomic mutations. **d** Correlation plots of gene expression between tissue and matched organoids for PCa case 61 (left) and case 62 (right). Pearson correlation coefficient, $r$: 0.795 and 0.858, respectively. For both correlations, $p$ value $< 2.2\mathrm{e}10^{-16}$.
**e** Results of PDO drug screen assay on three advanced PCa cases (P80, P82, P89) and two primary PCa cases (P133 and P134). Normalized viability $z$ scores are shown in the heatmap, with unsupervised hierarchical cluster analysis of the drugs. Asterisks indicate a significant reduction of viability compared to vehicle (*$p < 0.05$). Statistical significance was determined by two-tailed $t$-test for abiraterone (vehicle EtOH) and by one-way ANOVA with Dunnett correction for all remaining drugs (vehicle DMSO). Non-determined values are indicated in gray squares. Drug targets are indicated in the legend and reported by a colored-coded square below each drug.

modeled in vivo. Hypermutation and MSI are rare and sporadic features of PCa[47] that are associated with hereditary cancer predisposition. Despite the high tumor mutational burden (18–20 mut/MB), no deleterious alterations were detected in key MMR genes *MSH2, MSH6, MLH1,* or *PMS2* in PNPCa samples. Albeit rare, MSI-H features lacking a dMMR signature have been clinically observed[30,48]. Other molecular mechanisms, such as silencing of *MLH1* gene promoter or other epigenetic regulations may be responsible for the MSI-H status[49,50].

In PCa, high MSI is associated with poorly differentiated stage[51,52] ranging from 1% of primary tumor cases to up to 12% metastatic cases[53]. Among the patients with defined MSI-H/ dMMR molecular phenotype, ~50% respond to anti-PD-1/PD-L1 immunotherapy[54]. Given the lack of biomarkers for CRPC and the higher MSI prevalence in metastatic cases compared to primary cases, screening of patients for MSI status during initial diagnosis could determine whether anti-PD-1 treatment is the optimum treatment option. However, the lack of pressure of an immune system, common in both in vitro cultures and PDX passaging, could mask the pathophysiological role of this axis.

Organoid medium composition and overall cultural techniques were adapted from well-established and previously characterized studies[55,56], coherently with other recent research studies in the field[12,57]. However, in order to develop a translational assay, we eliminated extracellular matrix components to; increase drug availability, facilitate drug screening throughput and elimination of stromal cells. A drawback of this approach is that in our established methodology, organoids were generally used within a few passages. The preservation of high molecular correlation between PDX tissue and PDX-derived organoids was preferred over the opportunity to perform prolonged serial organoids propagation. Moreover, the predominant luminal phenotype observed in these PDX tumors was maintained in the organoid culture models.

We cross-validated drug responses in a complementary ex vivo assay on tissue slices, derived from the established PDXs based on our previously developed methodology[58]. The findings from both assays were similar, however organoid drug screenings are more amenable to standardization and can be performed routinely, in a shorter time frame and using limited amount of tissue from patient-derived material.

The implementation of PCa organoids in translational assays could facilitate the identification of effective therapies in non-responders to anti-androgens or chemotherapy, especially if repurposing drugs approved for other malignancies[59]. Differential responses to AR-blockers in LAPC9 and BM18/PNPCa organoids suggests that drug response correlates with individual tumor phenotypes. PNPCa was the only model showing sensitivity to EGFR-inhibitors: deregulation of EGFR signaling is found in a subset of PCa cases, however EGFR inhibitors have showed limited effectiveness[60,61]. Among the TKI inhibitors identified in

our screen, ponatinib emerged as broadly effective in metastatic PCa PDX as well as in PDOs. While sorafenib and sunitinib have been tested in phase II/III clinical trials for CRPC[62,63], ponatinib has not been yet investigated in PCa. Interestingly, both sorafenib and ponatinib were identified as CRPC candidate compounds based on a computational gene expression tool for drug identification[64]. and while its use in solid tumors is being currently investigated[65], it is approved mainly for patients with acute or chronic leukemia (PACE Trial[66]) and with acquired resistance to other TKIs[67]. When ultimately validated in vivo, ponatinib resulted in significant tumor growth inhibition and a very good tolerability for the duration of the experiment. These results corroborate the potential of organoids-based assays on drug screening applications in PCa.

We further developed a patient-derived organoid (PDOs)-based drug screening on a limited cohort of PCa patients, resulting in a descriptive, proof-of-principle study. PDOs recapitulated in most cases the genomic alteration of the matching tissues, both in metastatic and in primary lesions. In some cases, however we could not detect mutations in the PDOs or in both PDOs and matching tissues. This could be due to the low (or no) tumor content of the initial biopsy as well as to oncogenic mutations in regions outside those covered by our targeted DNA analysis. The developed PDO-based drug assay has an average duration of two weeks from initial organoid formation until readout, a time frame compatible with clinical decision-making. Individual inter-patient drug-sensitivity profiles, indicating the most effective compounds, even within the same drug subclass, are suggested by the results of the personalized drug screenings. Addressing intra-patient reproducibility was hampered due to the limited amount of material and of matching tumor-adjacent tissue available. In line with the high heterogeneity of PCa, the drug profiles correlated with disease stage, however they did not always match with the clinical history of the patient, highlighting the need to implement a near-patient assay. Our results highlight the applicability of patient-derived organoid drug screenings to predict clinical outcome[21] and their correlation with genomic and transcriptomic features of the primary tumor, as shown in recent studies for lung[68], gastrointestinal cancer[21], hepatocellular carcinoma[69], and ovarian cancer[70].

In summary, we presented a translational pipeline designed by cross-platform analysis of an early onset and treatment-naïve PCa xenograft model. The specific biologic and genetic landscape of this model may provide insights into tumor growth, metastasis, and drug resistance profile at an earlier stage of the disease. Comparison of its drug response profile with those of more advanced PCa PDXs allowed the generation of a highly translational tool for the evaluation of drug response in PDO, thus supporting a precision medicine approach to clinical decision-making.

## Methods

**Patient history**. The established PDX was originated from a patient who presented with primary PCa (Gleason 9) and underwent transurethral resection of the prostate (TUR-P) procedure. After 6 months, biopsy sampling was performed and the patient was diagnosed with a soft tissue metastasis; histopathology showed an infiltration of an adenocarcinoma from the prostate, PSA 91 ng/ml. Orchiectomy was performed directly after biopsy sampling, thus the tumor was androgen-dependent at the time of collection. No biochemical relapse was observed up to 18 months since the diagnosis (PSA < 1 ng/ml). All patients included in this study provided written informed consent (Cantonal Ethical approval KEK 06/03 and 2017-02295).

**Tumor sample preparation and xenograft surgery procedure**. Needle biopsy from soft tissue metastasis was collected in Dulbecco's MEM (Gibco, 61965-026) media containing primocin (InVivoGen). For xenograft implantation, needle biopsies were implanted subcutaneously in male, NOD-*scid* IL2Rgamma$^{null}$ (NSG) mouse, under aneasthesia (Domitor$^{®}$ 0.5 mg/kg, Dormicum 5 mg/kg, Fentanyl 0.05 mg/kg). Animal license BE 55/16 and BE 68/20. Weekly subcutaneous injections of testosterone propionate dissolved in castor oil (Sigma, 86541-5 G) were performed (2 mg per dosage, 25 G needle) starting 1 week after the surgery. For PDX passaging, serial subcutaneous implantations of tumor pieces into new recipients (NSG or CB17 SCID mice) was performed. Abiraterone acetate (Selleckchem, S2246) treatment was administered once daily (i.p.) for 5 days per week over a duration of 4 weeks at 0.5 mmol/kg/d (5% benzyl alcohol and followed by 95% safflower oil solution).

**DNA isolation from organoids and tissue samples**. For DNA extraction from organoids and tissue samples the DNeasy Blood and tissue kit (Qiagen, 69504) was used. DNA from FFPE material was extracted Maxwell$^{®}$ 16 LEV RNA FFPE Purification Kit (Promega, AS1260).

**RNA isolation and RT-qPCR**. RNA isolation from organoids was performed using the PicoPure Arcturus (Thermo Scientific, KIT0204) kit method. Tissue RNA was extracted using standard protocol of Qiazol (Qiagen) tissue lysis by TissueLyser (2 min, 20 Hz). Quality of RNA was assessed by Bioanalyzer (Agilent). RNA from FFPE material was extracted using the Maxwell$^{®}$ 16 LEV RNA FFPE Purification Kit (Promega, AS1260). Total RNA was used for cDNA synthesis using random primers and RNASe H-MML-V reverse transcriptase first-strand cDNA synthesis system (Promega AG, Dübendorf, Switzerland). For qPCR, cDNA (10 ng per reaction) was amplified in a CFX Real Time Detection system (Bio-Rad, Cressier, Switzerland) using SYBR Green Supermix reagent (Bio-Rad). Expression levels were normalized to the transcripts of *HPRT* and *ACTB*. Primer sequences are indicated in Supplementary Table 5.

**Tissue dissociation and organoid culture**. Tumor tissue was collected in Basis medium (Advanced DMEM F12 Serum Free medium (Thermo Fisher Scientific, 12634010) containing 10 mM Hepes (Thermo Fisher Scientific, 15630080), 2 mM GlutaMAX supplement (Thermo Fisher Scientific, 35050061), and 100 μg/ml Primocin (InVivoGen, ant-pm-1). After mechanical disruption the tissue was washed in Basis medium (220rcf, 5 min) and incubated in enzyme mix for tissue dissociation (collagenase type II enzyme mix (Gibco, 17101-015), 5 mg/ml dissolved in Basis medium, DNase: 15 μg/ml (Roche, 10104159001) and 10 μM Y-27632-HCl Rock inhibitor (Selleckchem, S1049). Enzyme mix volume was adjusted so that the tissue volume does not exceed 1/10 of the total volume and tissue was incubated at 37 °C for 1–2 h with mixing every 20 min. After digestion of large pieces was complete, the suspension was passed through 100 μm cell strainer (Falcon$^{®}$, VWR 734-0004) attached to a 50 ml Falcon tube. Using a syringe rubber the tissue was minced against the strainer and washed in 5 ml basic medium (220rcf, 5 min). Cell pellet was incubated in 5 ml precooled EC lysis buffer (150 mM NH$_4$Cl, 10 mM KHCO$_3$, 0.1 mM EDTA), incubated for 10 min, washed in equal volume of basis medium followed by centrifugation (220rcf, 5 min). Pellet was resuspended in 2–5 ml accutase™ (StemCell Technologies, 07920), depending on the sample amount; biopsies or tissue, and incubated for 10 min at room temperature. The cell suspension was passed through 40 μm pore strainer (Falcon$^{®}$, VWR 734-0004), and the strainer was washed by adding 2 ml of accutase on the strainer. Single cell suspension was counted to determine seeding density, washed in 5 ml of basis medium and spun down 220rcf, 5 min. Cell pellet was reconstituted in organoid medium and seeded in ultra low attachment (ULA) plates; e.g. 30,000 cells per well in 96-well plates with 100 μl media, 100,000 cells per well of 24-well plate with 750 μl media, 300,000–500,000 cells per well of 6-well plate (2 ml) (Corning, Costar #3471, 3473, 3474). Organoid culture media contains the following reagents: Basis medium containing 10 μM Y-27632-HCl (Selleckchem, S1049), 5% fetal calf serum (Gibco #10270-106, LOT 42G7277K), 1× B-27 supplement (Thermo Fisher Scientific, 17504044), 10 mM Nicotinamide (Sigma, N0636 100G), 500 ng/ml Rspondin (Peprotech, 120-38), 1.25 mM N-acetyl-cysteine (Sigma, A9165), 10 μM SB202190 (Selleckchem, S1077), 100 ng/ml Noggin (Peprotech, 250-38), 500 nM A83-01 (Tocris, 2939), 10 nM DHT (Fluka Chemica, 10300), 10 ng/ml Wnt3a (Peprotech, 315-20), 50 ng/ml HGF (Peprotech, 100-39), 50 ng/ml EGF (Peprotech, AF-100-15), 10 ng/ml FGF10 (Peprotech, 100-26), 1 ng/ml FGF2 (Peprotech, 100-18B), 1 μM PGE2 (Tocris, 2296). Media is prepared and kept at 4 °C for no longer than 7 days.

### Medium-throughput organoid drug screen at NEXUS personalized health technologies automation platform

*Compounds*. A drug library was compiled based on predicted activity against PCa (Selleck Chemicals, Houston, TX, USA) as 96-well format sample storage tubes with drugs in 10 mM concentration. Using a Tecan EVO 100 (Tecan AG, Männedorf, Switzerland), this drug library was aliquoted over 96-well plates (#651261, Greiner Bio-One, Kremsmünster, Austria) and further diluted to yield stock plates with a concentration of 10 mM, 1 mM, and 0.1 mM in DMSO (Sigma Aldrich, cat. D8418). After aliquotting, plates were sealed under argon gas using an Agilent PlateLoc (Agilent Technologies, Santa Clara CA, USA) with peelable aluminium heat-sealing foil (Agilent, cat. 24210-001) for 1 s at 170 °C. An overview of the purchased drugs and their known targets can be found in Supplementary Data 3. Control molecules enzalutamide, docetaxel, and doxorubicin were purchased at Sellechekchem (Lubio Science, Zürich, Switzerland, #S1250, #S1148, #S1208). Staurosporine was purchased at Toronto Research Chemicals (Toronto, Canada, #S685000).

*Automated drug screening with PCa organoids*. Automated screening procedures were performed at NEXUS Personalized Health Technologies (ETH Zürich, Zürich, Switzerland) using an automated screening platform (HighRes Biosolutions, Beverley, MA, USA). PCa organoids of BM18, LAPC9, or PNPCa origin were prepared and expanded from murine tumor tissue as for 5–7 days to allow organoid formation using Costar ultra-low attachment plates (#3471, Corning, New York, NY, USA). For the drug screens, organoids were dissociated into single cell suspension by both enzymatic (TrypLE incubation) and mechanical separation (22 G needle), counted and seeded in ULA 384 well plates at appropriate cell density for each tumor model; 3500 c/well (LAPC9, BM18) or 5000 c/well (PNPCa). Cells were seeded 25 μL per well in 384-well flat-bottom ultra-low attachment plates (#3827, Corning) using a BioTek EL406 with wide-bore 5 μL peristaltic pump tubing (BioTek Instruments Inc., Winooski, VT, USA). After cell seeding, plates were shaken for 2 min, incubated for 1 h at room temperature and subsequently transferred to a 37 °C incubator with 95% humidity and 5% CO$_2$. 48 h after cell seeding, 96-well plates (#651261, Greiner Bio-One) containing 1000-times concentrated compound stock solutions at differing concentrations (10, 1, and 0.1 mM) in DMSO were centrifuged at 250rcf for 10 s (HiG 5000, BioNex Solutions Inc., San Jose, CA, USA) and de-sealed using a Brooks Xpeel (Brooks Life Sciences, Chelmsford, MA, USA) and subsequently diluted 1:125 in culture medium and added to quadrant 1, 2, and 3 (respectively for 10, 1, and 0.1 mM stock plates) of deepwell 384-well plates (#781271, Greiner Bio-One). DMSO-stock solutions of control molecules (100% DMSO as negative control and as positive controls we included 1000-times concentrated docetaxel [30 μM], enzalutamide [6 mM], and doxorubicin [10 mM] for LAPC9 and BM18 organoids, or staurosporine [2.5 mM] instead of docetaxel for PNPCa organoids) were added to plate quadrant 4. An additional 1:1 dilution step was done prior to adding 20 μL of diluted drugs to the cell culture plates (containing 20 μL cell suspension after correction for evaporation). Compound dilutions were performed using automated liquid handling equipment (Tecan AG, Männedorf, Switzerland) in technical triplicate (LAPC9 and BM18) or technical quadruplicate (PNPCa). A Schematic representation of the compound dilution and addition procedure is shown in Supplementary Fig. 18a. After compound exposure, compound DMSO stock plates were sealed under argon gas using an Agilent PlateLoc as described above, and organoid culture plates were transferred back to the 37 °C incubator with a 95% humidity and 5% CO$_2$ atmosphere. 48 h after compound exposure, a CellTiter-Glo 3D assay (#G9682, Promega, Madison, WI, USA) was used to measure ATP levels as a proxy for cell viability. This assay is lytic and thus maximized readout from all cells composing large organoid structures. The cell viability readout was performed according to manufacturer's instructions using the automation equipment. Briefly, 40 μL room-temperature CellTiter-Glo 3D reagent was added per well to the assay plates using an automated liquid handler (Tecan AG). Plates were subsequently shaken for 5 min on a BioTek EL406 and incubated in a temperature-controlled incubator at 22 °C for 25 min. After incubation, luminescence was measured using a Tecan M1000 Pro plate reader (Tecan AG) with 1000 ms integration time. Results were collected as spreadsheets and coupled to plate layouts.

Data have been normalized using the median of the negative control conditions (DMSO 0.1%) and values have been log2 transformed (each plate with its own internal negative control). For the statistical analysis the different plates have been considered as replicates. *p*-value and FDR were calculated for all drugs after removing DMSO and no-treated control conditions).

*Material availability*. All unique materials are readily available upon request to the corresponding author.

**Reporting summary**. Further information on research design is available in the Nature Research Reporting Summary linked to this article.

## Data availability

All relevant data are available from the corresponding author. Sequencing data have been deposited at the European Genome-phenome Archive (EGA), under accession number EGAS00001004673 and EGAS00001004675. RNASeq related to Supplementary Fig. 10d

were deposited at the ENA (European Nucleotide Archive/Arrayexpress) under accession number E-MTAB-9656.

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

## Acknowledgements

We thank all patients who participated in our study, all involved clinical personnel and study nurses, in particular we would like to thank Anna-Katharina Herrmann and Sofia Bonné. We would like to acknowledge the Microscopy Facility of University of Bern, Viola Paradiso and Sina Maletti for the ion torrent sequencing. Swiss National Science Foundation 31003A_169352; 310030_189149; UniBe Initiator Grant 2016; KWF 2015_7599; Novartis 17B076; SOCIBP SPHN Driver Project (SAM).

## Author contributions

S.K. and F.L.M. designed experiments, acquired data, interpreted data, and wrote manuscript. A.B. performed extensive bioinformatics data analysis and revised the manuscript. M.K., M.D.M., E.Z., J.G., and I.K., performed additional experiments on clinical samples and animal experiments. A.G., M.B., A.V., J.-P.T., and M.R.D.F. performed bioinformatics data analysis and revised the manuscript. V.G. provided the pathological evaluation. D.K. acquired data and provided technical support. S.K., T.H.B., and C.U.S. generated the pipeline for FDA approved drug library tests, optimized the automation procedure for the PCa organoids, acquired data, and wrote the manuscript. K.E. performed sequencing experiments. A.S., C.K.Y.N., and S.P. performed bioinformatic data analysis. P.C.G. interpreted data and revised manuscript. M.S. provided clinical samples and revised manuscript. M.A.R. interpreted data and revised manuscript. G.N.T. provided clinical samples, interpreted data, and revised manuscript. M.K.-D.J. designed concept of the study and experiments, interpreted data, and wrote manuscript.

## Competing interests

The authors declare no competing interests.
