## [Peer Review File · Nature Communications]

Reviewers' comments:

Reviewer #1 (Expertise: Prostate cancer/PDX/organoids, drug response, Remarks to the Author):

This manuscript describes the establishment and molecular characterization of a PDX (PNPCA) from a castration sensitive prostate cancer metastasis, a drug screen of spheroids derived from PNPCA and two other previously established PDX models (one of which, LAPC9, also grows as a cell line), and the use of spheroid cultures derived from patient biopsies for therapeutic testing of a limited number of drugs.

Overall, there are no new biological conclusions of note and the novel technical aspects of the study are preliminary and require more in-depth data reporting and analysis. Specific comments are below for 1) PNPCA characterization, 2) therapeutics screening, and 3) patient biopsy spheroids.

1) PNPCA is shown to be castration sensitive and regrows after androgen add-back studies. This implies that selective tumor cells are long-term castration survivors. The novelty of this model would allow the analysis of such surviving cells, but this was not done. For example, genomic mutation analyses of "recurrent" tumor were not presented; the phenotype of surviving cells under androgen deprived conditions was not addressed; whether recurrent tumor had a modified androgen sensitivity was not presented.

With respect to the genotypic characterization, the authors emphasize the MSI-hi phenotype. However, it is unclear what the hypothesis is here. The conclusions regarding the population composition of the BRCA2 mutant phenotype (heterozygosity vs. loss of function) is not clearly addressed. Apparently MSH2/6 mutations are not evident. Is there any evidence of a homologous recombination deficiency signature? With respect to the phenotypic characterizations that were presented, for the transcriptomic analyses of PNPCA, BM18, and LAPC9 PDX/spheroid models, the correlation analysis comparing PDX and spheroids is a low bar. How do the various models compare in PCA plots? The constitutively expressed pathway analyses were relatively uninformative as the pathways described are generally observed in prostate cancers. Pathway hypothesis testing relative to identified pathway drivers, therapeutic response changes, etc. was not performed.

There are publicly-available cohorts of PC PDXs (which include MSI-hi and BRCA2mut genotypes), some of which demonstrate castration sensitivity, that have been molecularly characterized. Although PNPCA adds to the models available, it is not clear what the descriptive nature of this data adds to our knowledge.

2) For the therapeutic testing of established models (where cell numbers are not limiting), the presentation of the data is not robust. The relatively minimal difference between negative and positive controls (Figure S11) is cause for concern and suggests that dose response curves and AUC/IC50 comparisons should be shown as opposed to color-coded fold change. In addition, most of the drugs that scored (such as the anthracyclines, Figure 4) are commonly active in in vitro models. Although the authors conclude that ponatinib may be of interest for clinical use, there was no “ground truth” testing in vivo.

3) Given the historical difficulties culturing prostate cancer and the authors’ goal of utilizing the presented procedure to generate clinically useful information, it is incumbent upon the authors to thoroughly characterize those patient samples where there is sufficient material (Figure 5). Researchers in the field commonly have found that normal prostate basal cells from prostatectomies or contaminating epithelial cells from soft tissue biopsies outgrow prostate cancer cells. Since prostate cancer is quite heterogeneous in growth ability, it is arguable that patient samples that cannot be thoroughly characterized are likely to be useful for individualized therapy selection. To be believable, spheroids must be genomically characterized (whole or selected exomes and CNV) and compared to the tumor biopsy for depth and coverage. The spheroids also should be characterized with respect to lineage and tissue type. The present data is not fully interpretable.

Reviewer #2 (Expertise: Prostate cancer, genomics, Remarks to the Author):

This manuscript by Karkampouna and colleagues is a thorough and exhaustive analysis of a prostate cancer penile metastasis (PN met) biopsy grown as a patient-derived xenograft (PDX). The first ~60% of this manuscript focuses on the derivation and characterization of this tumor, demonstrating genomic and transcriptomic stability between the primary (TURP) tissue, the PN met biopsy, and the subsequent passages of the xenograft, both in the mouse and in Matrigel-free suspension “organoid” cultures. The authors’ characterization of this model includes the observation that the tumor is phenotypically similar to Lynch syndrome, showing microsatellite instability, which is unique for a prostate cancer model system. The final ~40% of this manuscript focuses on using this new organoid/PDX system as a drug screening platform, along with a series of other primary and advanced direct-to-organoid cultures.

In general, the data supporting the PNPcCa cell line and PDX/organoid derivation is strong. The authors rigorously challenged the stability of their PNPcCa model over multiple generations. Although there are a few concerns below, this superb characterization of this cell line will be of interest to many prostate cancer researchers looking for a model of MSI-high prostate cancer. However, the evidence supporting the utility of a rapid biopsy-to-organoid platform, featured in the last part of the paper (figure 5, supplementary fig 14, and Table 1) feels like a completely separate story. My comments below address the first 80% of the paper.

Major concerns:

1) The major premise of this work is that the authors' system for deriving new cell lines addresses the lack of PDX models for prostate cancer use. Notably missing from this work was any reference to the LuCaP series of PDXs generated at the University of Washington (USA) which indeed model androgen sensitive and insensitive (and neuroendocrine) prostate cancers. These lines, which now number greater than 30, are extensively characterized genomically and transcriptomically (Nguyen 2017, Kumar 2016) and were systematically developed into organoids with culture conditions adapted for each line (Beshiri 2018). The authors' criticism of prior work involving Matrigel (which includes LuCaP-derived organoids) puts an unmet burden of proof on the authors to demonstrate how their growth media outperforms the tried-and-true media developed by Clevers.

2) Dogmatically, these media formulations were developed for the propagation of luminal prostate cancer cells, which are notoriously challenging to grow (with a PDX take rate of 20-30% and an organoid take rate of <10%). Although the authors show by flow cytometry that they have enriched for populations that are CD44+, they also indicate that their CD49f high population is substantial. Normally, CD49f+ (or high) prostate cells are predominantly basal, not luminal cells, that will certainly form spheres in culture. However, since the PNPcCa line was derived from a metastasis and not a primary tumor, the most likely explanation is that the metastasis developed from a basal subtype tumor (see Zhao 2017). This should be addressed in the text since CD49f+ cells would be rare in most cultures.

3) The authors state in their results section that "PNPcCa-PDX tissue-derived organoids exhibit a luminal phenotype..." and then reference Supp Fig 4. This is clearly not the right evidence to support this data.

4) In Figure 2B, the authors describe this in the text as a comparison of growth kinetics. Since only one timepoint is shown, this is simply viability with and without DHT. Perhaps this data could be showed differently with respect the actual growth kinetics (time course viability) in the presence/absence of DHT?

5) Although I am very impressed with their authors' thorough use of transcriptome sequencing to characterize their lines, their correlations of gene expression between organoids and PDX tissue (fig 2C) should be higher. A thorough reading of the methods did not reveal any disambiguation or mouse-sequence removal step. This is critical, both for the DNA and RNA sequencing experiments as partially-mapping mouse reads with relatively high homology to the human sequence distort

differential gene expression analyses and mutation detection. A good tool for this is disambiguate but there are plenty to choose from.

6) In Figure 2D, the 'frequency of mutation' is shown to the right, which presumably is how frequently a mutation appears out of all 15 samples tested. The depiction of this data in supp fig 7 is far superior and should be moved to the main text. Some mutations are clonal and in all samples from a cell line. What about a mutation that only shows up in 1 out of 5? Or ones that only show up starting in the PDX? What evidence supports that mutation call? Is there additional focused re-sequencing to rule out any artifacts (especially after removal of mouse mapping reads from the sequence data)?

7) In general, there is very large number of mutations reported for the PNPcA cell line. Although this is possible because of the MSI-high (or hypermutator) phenotype, it would be useful to know if these mutations are all clustered in a single predominant clone or if they can be linked to 2 or more competing subclones in the population. Moreover, if more than one population is identified, how stable is that population over time from one generation to the next (serial PDX passages)? Tools like TITAN can estimate clonal prevalences while PyClone might be able to resolve up to 3 subclones. Analyses like these are critical for showing that the authors' modified methods at establishing and propagating these lines is preserving the heterogeneity of the tumor.

8) The authors use principal component analysis of their samples along with those from TCGA, depicted in Figure 9A/B. There are a number of issues with this, most notably that PCA will pick up variance between samples which here is undoubtedly of technical origin as these samples were not part of the original TCGA and were sequenced on a different platform. The authors' claim that the T1 clusters with CHD1-homdel tumors is not supported by this plot. The fact that the organoids and PDXs cluster so far apart, however, possibly indicates a lot of sample-to-sample variation (handling).

9) The relationship between MSI and PD-L1 function are not strongly supported by clinical trial data and the assertion made by the authors is not cited. MSI score is not predictive of response to immune checkpoint blockade, nor is high mutation count. PD-L1 positivity of a tumor is normally very strong staining in >5% of the tumor. This is not evident from the staining shown. Consequently, the tests for T cell activation are not thoroughly supported. Especially given that the source of T cells for the co-culture study were from healthy donors and not matched to the PNPcA cell donor, the lack of Treg induction by INF γ by PNPcA is inconclusive at best.

10) The evidence showing correlation between gene set enrichment and drug response is challenging to interpret. First, the data in Fig 4D suggests that every PNPcA were responsive to nearly every treatment. This is not entirely surprising, given that primary hormone sensitive PCa mets respond equally well to chemo and hormones. However, I'm not clear why the authors argue that baseline RNA-seq presented in figure 4C can be used to predict response. Of course the PNPcA will cluster more with BM18 based on ssGSEA values, as BM18 are androgen responsive. However, these genesets are only a subset of mSigDB and the KEGG pathways. What does unsupervised hierarchical clustering look like when all pathways are used? Given that these data apparently correlate, the challenges of assessing viability in vitro led the authors to perform ex vivo tissue slice drug treatments. This doesn't make very much sense – why can't the organoids be embedded and stained for KI67? Isn't the point of this platform to show its faster than working with tissue? The authors should clarify this point in the text.

MINOR:

- 1) In the introduction, the authors state that CHD1 and BRCA2 mutations are unique genomic features. These are quite common in metastatic prostate cancers. These are distinctive, perhaps, but not unique.
- 2) The ductal morphology of PNCa presented in Figure 1 and Supp Fig 1 should be addressed, as it is quite distinctive from the morphology of the TURP and PN met.
- 3) For all box plots, if $N < 10$ dots should be used instead. Error bars throughout the main and supplementary text are used inappropriately. For example, standard deviations or SEM bars are generally meaningless if $N < 10$. Instead, have the error bars depict the 95% confidence interval overlaid onto a dot plot.
- 4) The drug testing of PNCa appears to have been performed in tetraplicate (4 plates). This was buried in the methods but should be made clear in the figure legend or main text.
- 5) The authors state that the BM18 and PNCa models showed reduced expression of genes in the EMT, KRAS, JAK/STAT, WNT and NOTCH pathways. Compared to what? In Figure 4C, these are ssGSEA enrichment values that can't really be compared against "normal" samples except as a group. Do the authors mean reduced "enrichment" of these genes, or of the geneset. Perhaps a heatmap of the genes in question would eliminate confusion.

There are a number of serious issues with the data presented in Figure 5, the least of which is that it is not consistent with the message from the rest of the paper. Direct-to-culture approaches to culture primary human prostate cancer cells are going to strongly enrich for normal basal cells, which rapidly take over the population as an AR-positive androgen independent cell type quickly. Consequently, without any sorting or confirming lineage-specific immunohistochemistry or ISH of their organoids (i.e. NKX3.1, AR) directly on all embedded organoids, these cultures may be usable at best for up to 3 passages. The fact that two cases were shown in the main text (61 and 62) as concordant for gene expression/mutations but three different cases (80, 82, and 89) were used for drug testing raises the suspicion that the authors are not very successful with using this platform. How many cases were tried? What are the demographics of this cohort? Are there any similarities between successful and unsuccessful cases? The superficial discussion of gene expression and correlations between drug sensitivity from the primary or advanced PDOs makes me wonder whether this part of the manuscript is best kept for a separate write-up.

Point-by-point response to Reviewers' comments:

We would like to thank the Reviewers for their constructive comments and observations. Editing the manuscript according to the Reviewers' remarks resulted in a substantial enrichment of the scientific content of the manuscript, as well as in a more straightforward interpretation of the results. In order to properly address the revisions suggested by both Reviewers, we included in the current version of the manuscript additional original data, as well as revised analyses of the data previously submitted. We highlighted the changes to clearly mark the edits in the revised version of the manuscript. The updated results support and validate the original hypothesis, allowing us to maintain the order of the data as initially presented. However, the additional data resulted in the reorganization of some of the panels within the main figures and in additional supplementary figures and tables: in the current version of the manuscript as well as in this letter of reply to the Reviewers, we will be referring to the new order of the figures.

Reviewer 1:

R1Q1a

1) PNPCA is shown to be castration sensitive and regrows after androgen add-back studies. This implies that selective tumor cells are long-term castration survivors. The novelty of this model would allow the analysis of such surviving cells, but this was not done. For example, genomic mutation analyses of "recurrent" tumor were not presented; the phenotype of surviving cells under androgen deprived conditions was not addressed; whether recurrent tumor had a modified androgen sensitivity was not presented.

Edited items:

Edited main text [lines 155-160, 174-181], added **Fig.1F**, **Supp.Fig.2B**, **Supp.Fig.3B-C**, **Supp.Table 2**

Response:

To implement the suggestion of the Reviewer, we have added genomic sequencing on intact, castrated, and replaced tumors of the samples shown in **Fig.1D**, thus complementing the RNASeq analysis and IF staining of the surviving cells after castration, already included in the original manuscript. We performed genomic targeted sequencing on all 3 states (**Fig.1F and Supp.Table 2**), including the prolonged castration condition (40 weeks) which showed no spontaneous regrowth but responded to testosterone supplementation, achieving full regrowth in ~10 weeks (**Supp.Figures 2A and B**). The genotype of all states is largely consistent, meaning that the androgen-independent cells found in replaced conditions, and able to repopulate the tumor when in androgen rich conditions, did not undergo significant genomic evolution. However, we were able to detect some additional mutations after prolonged castration, as reported in the revised version of the text. The fact that the tumor did regrow shows that these surviving cells are not dependent on androgens for survival, but yet do depend on them for growth and proliferation.

Furthermore, we provided additional pathway analysis on the RNASeq data from the intact, castrated and testosterone-replaced conditions. No significantly differentially expressed genes could be found between the intact and the testosterone-replaced tumors, as supported by **Fig.1E** and **1G** (IF staining and PCA plot, respectively). We have addressed this in the text and by including the new **Supp.Fig.3B** (analysis of AR-related gene expression) and the new **Supp.Fig.3C** (pathway analysis of tumors between androgen-replaced and intact hosts). This latter analysis showed that tumors from androgen-replaced hosts had a downregulation in cell cycle, apoptosis, hypoxia related pathways (normalized enrichment score (NES) < -1), and upregulation of pathways linked to interferon response, protein secretion and androgen response (NES > 1).

R1Q1b

With respect to the genotypic characterization, the authors emphasize the MSI-hi phenotype. However, it is unclear what the hypothesis is here. The conclusions regarding the population composition of the BRCA2 mutant phenotype (heterozygosity vs. loss of function) is not clearly addressed. Apparently MSH2/6 mutations are not evident. Is there any evidence of a homologous recombination deficiency signature?

Edited items:

Edited main text [lines 289-291], edited **Supp.Fig.7**

Response:

BRCA2 mutation is heterozygous, with no evidence of HRD signature - based on large-scale transition (PMID 22933060). With the new mutational signature analysis following the mouse DNA clean up that we performed using Disambiguate (**Supp.Fig.7**, see R2Q5), there are basically no signs of the HRD signature on the mutational level.

R1Q1c

3)With respect to the phenotypic characterizations that were presented, for the transcriptomic analyses of PNPCCA, BM18, and LAPC9 PDX/spheroid models, the correlation analysis comparing PDX and spheroids is a low bar. How do the various models compare in PCA plots (Fig.2C)?

Edited items:

Edited main text [lines 205-207], added **Fig.2C**, edited **Fig.2D**

Response:

Our revised manuscript includes the suggested additional analysis comparing all the models in a principal component analysis plot (new **Fig.2C**). We also updated the correlation plots (**Fig.2D**) which are indicative of the transcriptomic concordance of organoids versus the tissue. Principal component analysis (PCA plot) indicated that the three PDX models were clearly distinct, and the corresponding PDX tissues and organoids were highly similar (**Fig.2C**).

R1Q1d

4)The constitutively expressed pathway analyses were relatively uninformative as the pathways described are generally observed in prostate cancers. Pathway hypothesis testing relative to identified pathway drivers, therapeutic response changes, etc. was not performed (Fig.4C). There are publicly-available cohorts of PC PDXs (which include MSI-hi and BRCA2mut genotypes), some of which demonstrate castration sensitivity, that have been molecularly characterized. Although PNPCCA adds to the models available, it is not clear what the descriptive nature of this data adds to our knowledge.

Edited items:

Edited main text [lines 264-274, 356-367], added **Fig.4D-E**, **Supp.Fig.10D**, **Supp.Fig.15** edited **Fig.4C** and **Supp.Fig.6**

Response:

We thank the Reviewer for the suggestions. We have updated **Fig.4C** and introduced substantial edits to the paragraph "*Organoid drug response to standard-of-care and repurposing of FDA approved compounds on a medium-throughput automated screen*" to clarify that we do not intend to provide a drug prediction analysis. The included and updated pathway analysis is rather to indicate that the results of the organoid screen have a molecular basis and that the organoids model is sufficiently representative of the original tissue. In regard to pathway hypothesis testing, we have updated the analysis included in the previous version of the manuscript. We have performed differential expression analysis followed by GSEA for cancer hallmarks (as defined in the Molecular Signatures Database), KEGG pathways as well as selected pathways to identify

concordance patterns among PDXs and PDX-derived organoids. The selected pathways were chosen based on mechanism of action /drug target for each effective compound according to PubChem.

As the Reviewer indicated, there are publicly available cohorts of characterized PDX and we have included further references to new studies in the revised manuscript. However, the majority of these cohorts are from advanced stage PCa (CRPC or NE), which have already acquired resistance traits due to patients' pretreatment with ADT or chemotherapeutics. In contrast, there is limited availability of early stage PCa PDXs recapitulating the natural course of early disease events. The PNPcCa model that we have established represents a hormone-naïve and treatment-naïve early metastasis model, which retains the molecular profile of its primary tumor (see **Fig.2E-F** and **Supp.Fig.5A, 8 and 9**). Thus it may serve as a reference point (early stage) for comparison with models of advanced PCa disease. We have now included a new analysis in the revised manuscript, comparing the gene expression profiles of PNPcCa, BM18, LAPC9 and a subset of LuCaP PDXs (**Sup.Fig.10D**).

R1Q2

2) For the therapeutic testing of established models (where cell numbers are not limiting), the presentation of the data is not robust. The relatively minimal difference between negative and positive controls (Figure S11) is cause for concern and suggests that dose response curves and AUC/IC50 comparisons should be shown as opposed to color-coded fold change. In addition, most of the drugs that scored (such as the anthracyclines, Figure 4) are commonly active in *in vitro* models. Although the authors conclude that ponatinib may be of interest for clinical use, there was no "ground truth" testing *in vivo*.

Edited items:

Edited main text [lines 332-336, 382-396], added **Fig.4D-E** and **Supp.Fig.15**, edited **Supp.Fig.12**

Response:

We thank the Reviewer for the crucial observation. We have reorganized **Supp.Fig.12** (substituting the former Supp.Fig.11) as follows: panels A-B show the PNPcCa pre-screen that justifies the timepoint choice for this model; panel C shows quality control data (data points distribution and Z-factor) for the controls of the actual PNPcCa screen and panel D (unchanged) shows the correlation plots for the assay replicates.

We used the Z-factor as a measure of statistical effect size, considering as acceptable a Z-factor value above 0, common practice for library screenings of compounds without a black/white effect. Dose response AUC or IC50 could not have been performed because we only included four doses per drug, while for a rigorous AUC/IC50 calculation a minimum of six doses is required. We chose a heatmap representation (normalised to the vehicle condition) as an easily readable graphical support to report the effects of all drugs and concentrations tested and to facilitate comparisons among samples (**Supp.Fig.13**, unchanged)

Following the Reviewer's suggestion, we evaluated the effect of ponatinib *in vivo* on the most aggressive tumor model (LAPC9, see updated **Fig.4D-E**, **Sup.Fig.15**). The results widely supported the findings of the *in vitro* screens: ponatinib significantly reduced tumor burden (tumor volume and weight at endpoint) and delayed tumor growth kinetic compared to the vehicle, thus in concordance with the effects observed *in vitro*.

R1Q3

3) Given the historical difficulties culturing prostate cancer and the authors' goal of utilizing the presented procedure to generate clinically useful information, it is incumbent upon the authors to thoroughly characterize those patient samples where there is sufficient material (Figure 5). Researchers in the field commonly have found that normal prostate basal cells from prostatectomies or contaminating epithelial cells from soft tissue biopsies outgrow prostate cancer cells. Since prostate cancer is quite heterogeneous in growth ability, it is arguable that patient samples that cannot be thoroughly characterized are likely to be useful for individualized

therapy selection. To be believable, spheroids must be genomically characterized (whole or selected exomes and CNV) and compared to the tumor biopsy for depth and coverage. The spheroids also should be characterized with respect to lineage and tissue type. The present data is not fully interpretable.

Edited items:

Edited main text [lines 196-198, 414-428 and 535-557], added **Fig.5B-C**, added **Supp.Fig.17**, edited **Fig.5E**

Response:

We thank the Reviewer for raising this important point. In our revised manuscript we have increased the sample size of our patient-derived organoid cohort and have substantially updated the corresponding Results and Discussion sections.

In order to thoroughly characterize the patient-derived organoids, we have performed DNA sequencing targeting genes recurrently mutated in PCa. We report the genomic profiles of patient-derived organoids compared to their original tissues and were able to confirm genomic matching mutations in 6/11 cases (**Fig.5B and C**). We have included eleven cases (the majority of which are from primary stage). Moreover, we have performed additional drug screens in cases of primary PCa organoids which were genomically confirmed as tumor organoids (**Fig.5B and E**).

Regarding the characterization of lineage type in the PDOs, we further included additional data (**Supp.Fig.17**) reporting the expression pattern of basal and luminal markers in PDOs and matched tissues at the RNA level, indicating the prevalent luminal phenotype of the low-passage organoids used in downstream assays/analysis.

Reviewer #2 (Expertise: Prostate cancer, genomics, Remarks to the Author):

This manuscript by Karkampouna and colleagues is a thorough and exhaustive analysis of a prostate cancer penile metastasis (PN met) biopsy grown as a patient-derived xenograft (PDX). The first ~60% of this manuscript focuses on the derivation and characterization of this tumor, demonstrating genomic and transcriptomic stability between the primary (TURP) tissue, the PN met biopsy, and the subsequent passages of the xenograft, both in the mouse and in Matrigel-free suspension “organoid” cultures. The authors’ characterization of this model includes the observation that the tumor is phenotypically similar to Lynch syndrome, showing microsatellite instability, which is unique for a prostate cancer model system. The final ~40% of this manuscript focuses on using this new organoid/PDX system as a drug screening platform, along with a series of other primary and advanced direct-to-organoid cultures.

In general, the data supporting the PNPcCa cell line and PDX/organoid derivation is strong. The authors rigorously challenged the stability of their PNPcCa model over multiple generations. Although there are a few concerns below, this superb characterization of this cell line will be of interest to many prostate cancer researchers looking for a model of MSI-high prostate cancer. However, the evidence supporting the utility of a rapid biopsy-to-organoid platform, featured in the last part of the paper (figure 5, supplementary fig 14, and Table 1) feels like a completely separate story. My comments below address the first 80% of the paper.

Major concerns:

R2Q1

1) The major premise of this work is that the authors’ system for deriving new cell lines addresses the lack of PDX models for prostate cancer use. Notably missing from this work was any reference to the LuCaP series of PDXs generated at the University of Washington (USA) which indeed model androgen sensitive and insensitive (and neuroendocrine) prostate cancers. These lines, which now number greater than 30, are extensively characterized genomically and transcriptomically (Nguyen 2017, Kumar 2016) and were systematically developed into organoids with culture conditions adapted for each line (Beshiri 2018). The authors’ criticism of prior work involving Matrigel (which includes LuCaP-derived organoids) puts an unmet burden of proof on the authors to demonstrate how their growth media outperforms the tried-and-true media developed by Clevers.

Edited items:

Edited main text [lines 93-97, 264-274 and 497-508], added **Supp.Fig.10D**

Response:

We agree with the Reviewer that the previous version of the manuscript lacked an adequate contextualization of the PDX investigated in this study in the broader landscape of available PDX models of PCa. We have updated the introduction section to provide additional context, and referenced additional PDX cohorts in the relevant Results and Discussion sections.

We have added a panel in **Supp.Fig.10D** showing cluster analysis of the transcriptomic signatures of the three PDX models investigated in this manuscript (PNPCa, BM18 and LAPC9) along with the androgen-sensitive, CRPC and NEPC PDXs from the LuCaP series. We chose to include data from the original LuCaP PDXs with reported androgen-sensitivity (Nguyen et al., 2017), excluding the castration-resistant sublines, which will be inherently different. As a technical note, since the samples presented in our study were characterized by whole transcriptome-based techniques (RNAseq), we could only include those PDXs within the LuCaP series for which we could generate a RNAseq-based transcriptional characterization: the transcriptomic data from the LuCaP PDX series referred in Reviewer’s 2, Question 1 was generated by microarray technology, limiting its usability when compared to RNAseq data due to confounding batch effects.

By performing hierarchical clustering of the RNASeq data from LuCaP, PNPcCa, BM18 and LAPC9 PDXs, we showed that the PNPcCa PDXs preferentially cluster with the more androgen-sensitive models (BM18 and the LuCaP 78), rather than with the more advanced models LAPC9 and LuCaP 147 and 35 or the neuroendocrine-like LuCaP 145.

We next addressed the MSI-related genomic features of the PNPcCa model in relation to the LuCaP PDX presented in **Supp.Fig.10D**. Among these models, only three were reported with hypermutation/MSI phenotype (Nguyen et al., 2017, Pritchard et al., 2014), including the CRPC LuCaP 147 PDX we used in the analysis. PNPcCa model clusters closely to the LuCaP 78 CRPC model, which does not contain *BRCA2* or MSI-related gene alterations. Instead LuCaP 23.1 (CRPC) and 145.2 (NEPC), despite carrying a heterozygous loss of *BRCA2*, do not cluster with the PNPcCa PDX model. So, in conclusion PNPcCa combines MSI-high and, *BRCA2* mutation, two genomic features frequently associated with more advanced PCa stages (CRPC, NEPC) Given the lack of available models recapitulating early stage PCa, especially from treatment naïve cases, the PNPcCa model can be used to model early events in PCa progression, steps towards androgen independence and acquisition of therapy resistance *in vivo*.

Prostate organoid methodologies have been indeed established primarily by the groups of M. Shen and H. Clevers respectively for mouse and human organoids, and have paved the way for new research directions in the field of PCa, including our study.

Moreover, the study by Beshiri *et al.* demonstrated that media composition requires adaptations for each organoid line. In this study, they have characterized extensively the LuCaP PDX-derived organoids and have improved the protocol for sustained proliferation (in particular successful for the advanced CRPC or NE cases).

Our ultimate goal was to generate a fast-turnover, near-patient drug screening pipeline on patient-derived organoids (PDOs). To achieve this, the elimination of Matrigel and implementation of suspension cultures not only proved to be optimal for short-term growth of our tumor organoids but also facilitated automated cell pipetting system for the medium throughput screens. The organoid medium we employed is based on the medium defined by Gao *et al.* (2014) with minor modifications, to achieve short-term organoid growth, not indefinite passaging. In this system, the organoids do maintain the key genotypic and phenotypic properties of their originating tumor tissues and thus can be used as translational tools to provide clinically relevant information on drug sensitivity. We have updated the Discussion with references to the cohort of LuCaP series of PDXs and organoid studies.

R2Q2

2) Dogmatically, these media formulations were developed for the propagation of luminal prostate cancer cells, which are notoriously challenging to grow (with a PDX take rate of 20-30% and an organoid take rate of <10%). Although the authors show by flow cytometry that they have enriched for populations that are CD44+, they also indicate that their CD49f high population is substantial. Normally, CD49f+ (or high) prostate cells are predominantly basal, not luminal cells, that will certainly form spheres in culture. However, since the PNPcCa line was derived from a metastasis and not a primary tumor, the most likely explanation is that the metastasis developed from a basal subtype tumor (see Zhao 2017). This should be addressed in the text since CD49f+ cells would be rare in most cultures.

Edited items:

Edited main text [lines 196-198 and 497-508], edited **Supp.Fig.5A**

Response:

We are familiar with the molecular subtyping work by Zhao et al.,2017, in which they observed correlation of luminal /basal subtype with clinical outcome and reported that the basal lineage CD49f signature gene is enriched more in basal tumors compared to luminal tumors. The PNPCa PDX is predominantly luminal, both morphologically and molecularly, as we have extensively characterised. However, the protein expression of CD49f is indicative of heterogeneous subpopulations which could be present. At the molecular level, gene expression profiling by RNASeq of these organoids showed on average that they are predominantly luminal as the expression of basal signature genes is lower compared to luminal genes (*NKX3.1*, *KRT18*, *AR*); see updated **Supp.Fig.5A**). Expression of *KRT5* and *KRT6A* (basal genes) is minimal or not detected. The above is in line with the luminal cell type composition observed in the organoid cultures, which were positive for AR, PSA, NKX3.1 and CK8 luminal markers (Fig.2A). The text was modified to include this information.

R2Q3

3) The authors state in their results section that “PNPCa-PDX tissue-derived organoids exhibit a luminal phenotype...” and then reference Supp Fig 4. This is clearly not the right evidence to support this data.

Edited items:

Edited main text [lines 196-198 and 506-508], added **Supp.Fig.5**

Response:

We thank the Reviewer for highlighting this point. We have added RNASeq data on luminal/basal markers in **Supp.Fig.5A**. and amended the reference to the **Supp.Fig.5B** where relevant in the main text.

R2Q4

4) In Figure 2B, the authors describe this in the text as a comparison of growth kinetics. Since only one timepoint is shown, this is simply viability with and without DHT. Perhaps this data could be showed differently with respect the actual growth kinetics (time course viability) in the presence/absence of DHT?

Edited items:

Edited main text [lines 200-202]

Response:

Correct, this is not a growth kinetic assay but viability assay of different organoid models in response to DHT, we have adjusted the text accordingly.

R2Q5

5) Although I am very impressed with their authors' thorough use of transcriptome sequencing to characterize their lines, their correlations of gene expression between organoids and PDX tissue (fig 2C) should be higher. A thorough reading of the methods did not reveal any **disambiguation or mouse-sequence removal step**. This is critical, both for the DNA and RNA sequencing experiments as partially-mapping mouse reads with relatively high homology to the human sequence distort differential gene expression analyses and mutation detection. A good tool for this is disambiguate but there are plenty to choose from.

Edited items:

Edited main text [lines 205-207], edited Supplementary methods, added **Fig.2C**, updated **Fig.2D**, **2E**, **3D**, **3E**. Updated **Supp.Fig.7**, **8B**, **9**, **11A** and **SI Table 3**

Response:

We thank the Reviewer for this observation. For the RNASeq data, the alignment was actually performed against a hybrid human and mouse genome and not only against the human genome as written in the methods in the original submission. The relevant methods section has been corrected accordingly. Regarding the correlations between PDXs and organoids (**Fig.2D**) a range of r between 0.89-0.92 indicates a good although not completely matching transcriptomic profile, which is reasonable given that we compare the very different settings of *in vitro* and *in vivo* models. We have also included a principal component analysis plot comparing all the different models together, which shows a high degree of similarity between the PDXs and the corresponding organoids (**Fig.2C**).

For the WES data, we have realigned all data (PNPCa, BM18, LAPC9) to the mouse genome and used 'disambiguate', as suggested by the Reviewer, to extract human-specific reads. We removed previous variant calls that were no longer evident in the human-specific reads. In addition, for BM18 and LAPC9 without matched germline, we added an additional filter by removing probable polymorphisms as listed in the GnomAD population database. In the original version of the manuscript we had used only the ExAC population dataset whereas now more probable polymorphisms have been removed, further increasing the enrichment for likely genuine somatic mutations in the presented samples. For the Ion Torrent data presented in **Fig.1F** and **Supp.Fig.2B**, the data were aligned simultaneously to human and mouse genomes. As a result of these changes, many figures and data have been updated, but the previous conclusions essentially remain the same. The methods description and the following figures have been revised.

R2Q6

6) In Figure 2D, the 'frequency of mutation' is shown to the right, which presumably is how frequently a mutation appears out of all 15 samples tested. The depiction of this data in supp fig 7 is far superior and should be moved to the main text. Some mutations are clonal and in all samples from a cell line. What about a mutation that only shows up in 1 out of 5? Or ones that only show up starting in the PDX? What evidence supports that mutation call? Is there additional focused re-sequencing to rule out any artifacts (especially after removal of mouse mapping reads from the sequence data)?

Edited items:

Edited main text [lines 221-226 and 246-252], added **Fig.1F**, **Fig.2F**, updated **Fig.2E**, **Supp.Fig.7**, **Supp.Fig.9**

Response:

We thank the Reviewer for the recommendations. We have removed the frequency of mutation plots, which were not highly informative, and we have updated **Fig.2E** (formerly Fig.2D) after the removal of the mouse reads as well as the cancer cell fraction heatmaps in **Supp.Fig.7** and **Supp.Fig.9**. Given the space constraints, we have kept **Fig.2E** in the main text, along with the addition of subclonal analysis in **Fig.2F**, facilitating the comparison between and the distribution of the different mutations in the PNPCa models.

During our WES analysis, we first called mutations in each individual sample *de novo*, and then we re-genotyped the union of all called mutations to identify additional mutations that were not called but may be present. This was performed in particular due to the lower tumor purity of T1. Due to the low tumor purity of T1, in fact, approximately half of the reported mutations in T1 were re-genotyped this way. This would suggest that a lot of the mutations that were detected *de novo* in P2/P3/P4/Org2 were indeed present in T1 but were below (*de novo*) detection levels. One such example is the *BRCA2* p.Asn863fs mutation, which is also detected by Ion Torrent sequencing in all the other PNPCa samples (**Fig.1F**). This suggests that there are likely many more such mutations that were not captured in the T1 sequencing but are likely to be present. Similarly, of the mutations that were called *de novo* in only 1 of 5 samples, most of them could also be detected but not called *de novo* in at least one other sample, suggesting that there are likely many more such mutations.

R2Q7

7) In general, there is very large number of mutations reported for the PNPCa cell line. Although this is possible because of the MSI-high (or hypermutator) phenotype, it would be useful to know of these mutations are all clustered in a single predominant clone or if they can be linked to 2 or more competing subclones in the population. Moreover, if more than one population is identified, how stable is that population over time from one generation to the next (serial PDX passages)? Tools like **TITAN** can estimate clonal prevalences while PyClone might be able to resolve up to 3 subclones. Analyses like these are critical for showing that the authors' modified methods at establishing and propagating these lines is preserving the heterogeneity of the tumor.

Edited items:

Edited main text [lines 246-252], added **Fig.2F**

Response:

We have performed clonal analysis using PyClone (**Fig.2F**). The data indicate that these mutations cluster in multiple subclones, with tumor heterogeneity stably preserved in the samples analyzed, as most variants were roughly equally abundant in all samples (including the primary T1), which corroborates results shown in **Supp.Figures 7 and 9**.

R2Q8

8) The authors use principal component analysis of their samples along with those from TCGA, depicted in Figure 9A/B. There are a number of issues with this, most notably that PCA will pick up variance between samples which here is undoubtedly of technical origin as these samples were not part of the original TCGA and were sequenced on a different platform. The authors' claim that the T1 clusters with CHD1-homdel tumors is not supported by this plot. The fact that the organoids and PDXs cluster so far apart, however, possibly indicates a lot of sample-to-sample variation (handling).

Response:

We agree with the Reviewer that technical batch effects between datasets or sequencing centers could confound PCA. However, our analysis suggests that inter-individual variance is higher than technical, batch-dependent variance in this context which, albeit present, is not predominant in the two principal components (PC) plotted in the PCA graphs included in **Supp.Fig.10**. To more robustly address this point, we integrated the TCGA dataset with two additional independent datasets of primary prostate cancer (blue and green datapoints in the plot below), resulting in an equivalent representation. We did not include these two datasets in the manuscript because of their lack of clinical and mutational information on key genes (FOXA1, SPOP, etc.) investigated.

Although we cannot exclude that PCA positioning of PDX models may be partially affected by the different biological environment in which the tumor is growing, a PCA plot generated by excluding the custom samples (PNPCa-related samples, indicated in yellow) results in a similar representation of the primary tumors.

We also agree that unbiased PCA positioning is not clearly positioning the custom samples among CHD1-deleted tumors. To this end, a single-sample gene-set enrichment analysis (ssGSEA) approach allows to better capture the transcriptional similarity among the custom samples and CHD1-deleted prostate cancer tumors. This analysis is included in **Supp.Fig.10C**; a box plot generated with the same data but illustrating the specific CHD1-homdel induced and repressed transcriptional signatures is reported below.

R2Q9

9) The relationship between MSI and PD-L1 function are not strongly supported by clinical trial data and the assertion made by the authors is not cited. MSI score is not predictive of response to immune checkpoint blockade, nor is high mutation count. PD-L1 positivity of a tumor is normally very strong staining in >5% of the tumor. This is not evident from the staining shown. Consequently, the tests for T cell activation are not thoroughly supported. Especially given that the source of T cells for the co-culture study were from healthy donors and not matched to the PNPcCa cell donor, the lack of Treg induction by INF γ by PNPcCa is inconclusive at best.

Edited items:

Edited main text [lines 296-300, 315-322, 484-496]

Response:

We thank the Reviewer for bringing this important aspect to our attention. We have edited the main text in the relevant Results and Discussion sections, adding further recent clinical studies that focused on link between anti-PD-L1/PD-1 therapy response and MSI status in prostate cancer. Given the MSI status of the PNPcCa PDX, we set to explore its immunological background in terms of PD-L1 expression and functionality. As of note, despite the increasing interest towards immune checkpoint inhibitors, a consensus for a reliable staining of these markers, in particular for PD-L1, is still debated in the field. In order to corroborate the weak levels of expression of PD-L1 in PNPcCa tissues (**Fig.3F**), we further evaluated a gene expression panel of immune checkpoint inhibitors (**Fig.3G** and **Supp.Fig.11C-E**). The immunological assays proposed in **Fig.3H-J** are intended as complementary to **Fig.3F** and show that, despite the upregulation of PD-L1 when stimulated with IFN-gamma *in vitro*, PNPcCa organoids do not show significant immunomodulatory functions. In conclusion, we are in line with the point raised by the Reviewer and we are aware that the functional immune assays reported in **Fig.3H-J** are by no means a conclusive proof of the link between MSI status and immunological properties of PNPcCa organoids.

R2Q10

10) The evidence showing correlation between gene set enrichment and drug response is challenging to interpret. First, the data in Fig 4D suggests that every PNPcCa were responsive to nearly every treatment. This is not entirely surprising, given that primary hormone sensitive PCa mets respond equally well to chemo

and hormones. However, I'm not clear why the authors argue that baseline RNA-seq presented in figure 4C can be used to predict response. Of course the PNPcCa will cluster more with BM18 based on ssGSEA values, as BM18 are androgen responsive. However, these genesets are only a subset of mSigDB and the KEGG pathways. What does unsupervised hierarchical clustering look like when all pathways are used? Given that these data apparently correlate, the challenges of assessing viability *in vitro* led the authors to perform *ex vivo* tissue slice drug treatments. This doesn't make very much sense – why can't the organoids be embedded and stained for KI67? Isn't the point of this platform to show its faster than working with tissue? The authors should clarify this point in the text.

Edited items:

Edited main text [lines 210-220, 356-367 and 511-514], added **Fig.2C**, edited **Fig.4C-E**, **Supp.Fig.6** and **Supp.Fig.13**

Response:

We thank the Reviewer for bringing these points to our attention. Regarding the differential drug responses of PNPcCa organoids compared to the other models, we edited the text to better clarify that the organoids drug screen consisted of 74 distinct drug candidates. Despite showing response to more candidate hits than any other model (**Fig.4B**), PNPcCa organoids were significantly sensitive to only 14 of the total 74 tested compounds; a heatmap listing the effects of all the tested compounds is reported in **Supp.Fig.13**.

We have revised both **Fig.4C** and **Supp.Fig.6**, showing side-by-side the pathway enrichment analysis of PDX tumor tissues and their derived organoids. We also performed a more exhaustive pathway analysis using the entire HALLMARK and all C2 KEGG datasets, further including unsupervised hierarchical clustering analysis of both the pathways and the samples, as the Reviewer suggested (**Supp.Fig.6**). Differential expression analysis of all samples versus control tissue (non-carcinoma, N1) was followed by GSEA of HALLMARK and C2 KEGG, to allow comparisons among the different samples and showed clustering per PDX model (tumor tissue with the organoids of each particular model). The updated heatmap reports pathways with a significant enrichment score (FDR < 0.05, **Fig.4C**). Although one might have expected PNPcCa would be more similar to BM18 given that both models are androgen-sensitive, a principal component analysis (**Fig.2C**) of the top 1000 most variable genes, would suggest this is not the case.

Correlating transcriptomic profile and drug responses would be indeed speculative at best with the data provided in the manuscript and was outside the scope of presented results. Therefore, in the revised manuscript, we substantially edited the text referring to **Fig.4B-C**, better clarifying that the pathway analysis was included to show that, despite the drug assay was performed *in vitro* on organoids, the molecular pathways most relevant to the screened hits exhibit a similar enrichment pattern in both organoids and their originating PDX tissues. The *ex vivo* validation assay and, since this revised version of the manuscript, the *in vivo* experiment (**Fig.4D-E**) further support the results obtained by the organoids drug screening. A relevant goal of this study, as illustrated in **Fig.5**, is to establish a time-effective pipeline to screen early-passage, patient-derived organoids with a defined panel of drugs, shortlisted via the presented medium-throughput screening. The use of a molecular assay to measure viability is aimed at increasing the consistency, sensitivity and readability of drug effectiveness on organoids.

MINOR:

1) In the introduction, the authors state that CHD1 and BRCA2 mutations are unique genomic features. These are quite common in metastatic prostate cancers. These are distinctive, perhaps, but not unique.

Edited items:

Edited main text [lines 111-113]

Response:

We have rephrased the text in the introduction according to the Reviewer's comment.

2) The ductal morphology of PNPcCa presented in Figure 1 and Supp Fig 1 should be addressed, as it is quite distinctive from the morphology of the TURP and PN met.

Edited items:

Edited main text [lines 127-129]

Response:

The primary TURP tissue contains ductal structures with heterogeneous morphology, while the PDX tumors have a very distinctive ductal morphology. Such alteration might be attributed to specific selection of cancer cell subpopulations which are either present in the TURP but absent in the PNmet, or due to selection and propagation of specific clones during the PDX evolution (**Fig.2F**). We have addressed this morphological aspect in the Results section to reflect this point.

3) For all box plots, if N<10 dots should be used instead. Error bars throughout the main and supplementary text are used inappropriately. For example, standard deviations or SEM bars are generally meaningless if N<10. Instead, have the error bars depict the 95% confidence interval overlaid onto a dot plot.

Edited items:

Edited **Fig.2B, Fig.3G-J, Supp.Fig.11E, Supp.Fig.14B**

Response:

All the relevant data now include the individual datapoints, with error bars indicating the standard deviation. In **Sup.Fig.11C and D, Sup.Fig.4A and 17**, we show marker expression from RNASeq data for which N=1.

4) The drug testing of PNPcCa appears to have been performed in tetraplicate (4 plates). This was buried in the methods but should be made clear in the figure legend or main text.

Edited items:

Edited main text [lines 332-336]

Response:

We have emphasized this information in the figure legends and in Results description of **Fig.4C** and **Sup.Fig.13**.

5) The authors state that the BM18 and PNPcCa models showed reduced expression of genes in the EMT, KRAS, JAK/STAT, WNT and NOTCH pathways. Compared to what? In Figure 4C, these are ssGSEA enrichment values that can't really be compared against "normal" samples except as a group. Do the authors mean reduced "enrichment" of these genes, or of the geneset. Perhaps a heatmap of the genes in question would eliminate confusion.

Edited items:

Edited main text [lines 356-367] and **Fig.4C**

Response: Acknowledging the Reviewer's comment and to eliminate further misinterpretations of the pathway analysis, we have adopted a different approach for **Fig.4C** performing differential expression analysis of each and all of the samples over a non-carcinoma control tissue (N1), followed by GSEA of HALLMARKS and 2 KEGG (C2 subset) genesets. We then compare side by side the PDX tissues and their derived organoids, given that the drug screenings were performed on the organoid models. The updated heatmaps highlight pathways with significant normalized enrichment score (NES > |1| and FDR < 0.05) as a way to describe the biologic processes modulated in the PDXs and Organoids, compared to control tissue.

Reviewer's additional comment

There are a number of serious issues with the data presented in Figure 5, the least of which is that it is not consistent with the message from the rest of the paper. Direct-to-culture approaches to culture primary human prostate cancer cells are going to strongly enrich for normal basal cells, which rapidly take over the population as an AR-positive androgen independent cell type quickly. Consequently, without any sorting or confirming lineage-specific immunohistochemistry or ISH of their organoids (i.e. NKX3.1, AR) directly on all embedded organoids, these cultures may be usable at best for up to 3 passages. The fact that two cases were shown in the main text (61 and 62) as concordant for gene expression/mutations but three different cases (80, 82, and 89) were used for drug testing raises the suspicion that the authors are not very successful with using this platform. How many cases were tried? What are the demographics of this cohort? Are there any similarities between successful and unsuccessful cases? The superficial discussion of gene expression and correlations between drug sensitivity from the primary or advanced PDOs makes me wonder whether this part of the manuscript is best kept for a separate write-up.

Edited items:

Edited main text [lines 414-428, 432-434 and 535-557]

Response:

In our revised manuscript, we have addressed the Reviewers' concerns about sample size and quality. We have increased the sample size of the PDOs presented in the study (**Fig.5C, E**). We included genomic sequencing of all the PDO cases presented, the majority of which is from primary PCa. Unfortunately, due to the very limited amount of patient-derived material available for each patient, we cannot perform an extensive characterization on all PDO cultures. However, we have now included **Fig.5B** that summarizes the success rate of PDO generation. Overall, we generated PCa tumor organoids harboring the same mutations as the original tissue in about 75% of cases. We have now included **SI Table 6** providing the relevant demographic and clinical data of the patient's samples included in the study.

The Reviewers' additional concerns about normal and/or basal cells overtaking PDO cultures are a shared concern in the organoids field. Presence of basal cells in PDOs cannot be excluded of course and in fact organoids must recapitulate the basal/luminal heterogeneity of the tissue: we provide evidence of this by showing that the transcriptomic correlation of organoids and tissue is very high and by the inclusion of **Supp.Fig.17** showing the prevalent expression of luminal markers in PDOs and matched tissues. However, their overgrowth in the organoid cultures can be excluded by the fact that somatic mutations, common to the tissue, are detected (**Fig.5C**). We have also updated the discussion to better address this concern, specifying that prolonged serial passaging of PCa organoids is beyond our aims and that our culture technique, while moving from methods well established in the field, is aimed at the development of near-patient assays.

The results on PDO recapitulate the evidence collected for PDX samples, presented in the previous figures and showing a higher RNASeq expression of luminal markers compared to basal markers in both PDX tissues and organoids (**Sup.Fig.4**).

REVIEWERS' COMMENTS

Reviewer #1 (Remarks to the Author):

The authors are commended for having addressed many of the previous criticisms. However, a mutational/molecular explanation for the genetic instability of PNPcA was not resolved would increase considerably the potential interest and utility for this model. The manuscript in general incorporates a variety of data that is descriptive and doesn't lead to firm conclusions (other than prostate cancer models are heterogeneous). In addition, the intra-patient reproducibility (feasible with prostatectomy and resected metastatic samples) of biopsy-derived spheroid drug responses and a comparison to normal prostate are missing validations for "proof of principle."

Reviewer #2 (Remarks to the Author):

In their revision, the authors have impressively addressed all of the issues raised during the previous round of review. The revised manuscript incorporates a series of new analyses that further strengthen their paper. The updated figures and supplementary figures provide greater support to the conclusions of their initial draft. All of the major and minor issues I raised were appropriately handled.

My only minor request is that if space allows, the authors move as much of the supplementary methods/materials to the main text, especially if they do not count against the word limit.

Point-by-point response to Reviewers' comments:

Reviewer #1 (Remarks to the Author):

The authors are commended for having addressed many of the previous criticisms. However, a mutational/molecular explanation for the genetic instability of PNPcCa was not resolved would increase considerably the potential interest and utility for this model. The manuscript in general incorporates a variety of data that is descriptive and doesn't lead to firm conclusions (other than prostate cancer models are heterogeneous).

Response:

We thank the Reviewer for the constructive comments. The microsatellite instability of the PNPcCa tumors, was confirmed by three distinct approaches; the oncogenic mutational signature analysis, the MSIsensor algorithm and the Bethesda NIH MSI gene panel, which confirmed the hypermutation profile of this tumor (18-20 Mut/MB, high TMB ≥ 10) and its MSI-high features. Genomic inactivation of mismatch repair genes is a frequent molecular feature that contributes to MSI. However, there were no detectable deleterious alterations in the key MMR genes *MSH2*, *MSH6*, *MLH1* or *PMS2* in PNPcCa samples. To provide an alternative explanation, we assessed aberrations in genes of the homologous recombination pathway, however none were detected with the exception of the *BRCA2* somatic mutation (Results section, lines 291-293). Other studies have reported patient cases with evidence of MSI and yet undetectable MMR gene mutations (Graham *et al.*, 2020, doi: [10.1371/journal.pone.0233260](https://doi.org/10.1371/journal.pone.0233260), Abida *et al.*, 2019 Supplementary e-table 1 and e-table 2 DOI: [10.1001/jamaoncol.2018.5801](https://doi.org/10.1001/jamaoncol.2018.5801)) and were reported in the updated discussion section [lines 487-499]. A study from Abida *et al.* reported that out of 32 MSI-H/dMMR prostate cancer patients, 78% has no germline mutation in a MMR-associated gene. Therefore, the frequency of patients who had a confirmed pathogenic or likely pathogenic germline mutation in an MMR-associated gene is rather low (22%), indicating that MMR aberrations did not drive the pathogenesis of these tumors. Other underlying mechanisms may occur, such as epigenetic silencing of MMR-gene promoter without the gene mutation, as reported for the hypermethylation of *MLH1* promoter in sporadic, non-familial cases of colorectal cancer (PMID: 9041175, PMID: 11870540). Whether such epigenetic events take place in PCa tumors requires further investigations that are beyond the scope of the current study.

In addition, the intra-patient reproducibility (feasible with prostatectomy and resected metastatic samples) of biopsy-derived spheroid drug responses and a comparison to normal prostate are missing validations for “proof of principle.”

Given the extensive PDX and PDX-derived organoid characterisation, in terms of genomic and transcriptomic profiles as well as drug responses *in vitro* and *in vivo* drug validation, our proof of principle study aimed at: i) the generation of prostate cancer patient derived organoids ii) the characterisation of their genomic landscape and that of the matching tumor and iii) the assessment of *in vitro* drug response. Our future studies, now aim to implement the developed models towards a co-clinical trial.

To acknowledge the limitations of our study we further specified in the Discussion that i) our study was based on a limited cohort of PCa patients, resulting in a descriptive, proof-of-principle study and ii) intra-patient reproducibility was hampered due to the limited amount of material and of matching tumor-adjacent tissue available.

Reviewer #2 (Remarks to the Author):

In their revision, the authors have impressively addressed all of the issues raised during the previous round of review. The revised manuscript incorporates a series of new analyses that further strengthen their paper. The updated figures and supplementary figures provide greater support to the conclusions of their initial draft. All of the major and minor issues I raised were appropriately handled.

My only minor request is that if space allows, the authors move as much of the supplementary methods/materials to the main text, especially if they do not count against the word limit.

Response:

We thank the Reviewer for the positive comments and careful review which helped improve the manuscript.